# Localist Topographic Expert Routing: A Barrel Cortex-Inspired Modular Network for Sensorimotor Processing

**Tianfang Zhu**[1], **Dongli Hu**[1], **Jiandong Zhou**[1], **Kai Du**[2*], **Anan Li**[1,3] *

[1]Wuhan National Laboratory for Optoelectronics,
Huazhong University of Science and Technology
[2]Psychological and cognitive sciences, Tsinghua University
[3]School of Biomedical Engineering, Hainan University
[1]{funfunfun,hudongli,jiandongzhou,aali}@hust.edu.cn    [2]kai_du@tsinghua.edu.cn

## Abstract

Biological sensorimotor systems process information through spatially organized, functionally specialized modules. A canonical example is the rodent barrel cortex, in which each vibrissa (whisker) projects to a dedicated cortical column, forming a precise somatotopic map. This anatomical organization stands in stark contrast to the architectures of most artificial neural networks, which are typically monolithic or rely on expert-isolated mixture-of-experts (MoE) mechanisms. In this work, we introduce a brain-inspired modular architecture that treats the barrel cortex as a biologically constrained instantiation of an expert system. Each module (or "expert") corresponds to a cortical column composed of multiple neuron subtypes spanning vertical cortical layers. Sensory signals are routed exclusively to their corresponding columns, with inter-column communication restricted to local neighbors via a sparse gating mechanism. Despite these anatomical constraints, our model achieves competitive, state-of-the-art performance on challenging 3D tactile object classification benchmarks. Columnar parameter sharing provides inherent regularization, enabling 97% parameter reduction with improved training stability. Notably, constrained localist routing suppresses inter-module activity correlations, mirroring the barrel cortex's lateral inhibition for sensory differentiation, while suggesting MoE's potential to reduce expert redundancy through collaborative constraints. These results suggest how cortical principles of localist expert routing and topographic organization could potentially be translated into machine learning architectures. Code is available at `https://github.com/fun0515/MultiBarrelModel`.

## 1 Introduction

One of the hallmarks of biological intelligence is its use of modular, topographically structured systems to process sensorimotor information [21, 45, 20]. In rodents, the barrel cortex [37, 44] epitomizes this principle: each whisker maps one-to-one onto a dedicated cortical column, enabling localized, efficient processing of tactile input (Fig. 1A). This precise sensor-to-column mapping supports robust spatial discrimination and energy-efficient computation. Extensive neuroscientific studies have characterized both the vertical microcircuitry [12, 36, 48, 14] within each column—organized across layer 2 (L2) through L6—and the horizontal connections that support lateral integration between neighboring columns [32, 12, 23]. However, the implications of this biologically modular and anatomically constrained design for artificial intelligence remain underexplored.

---

*Corresponding authors.

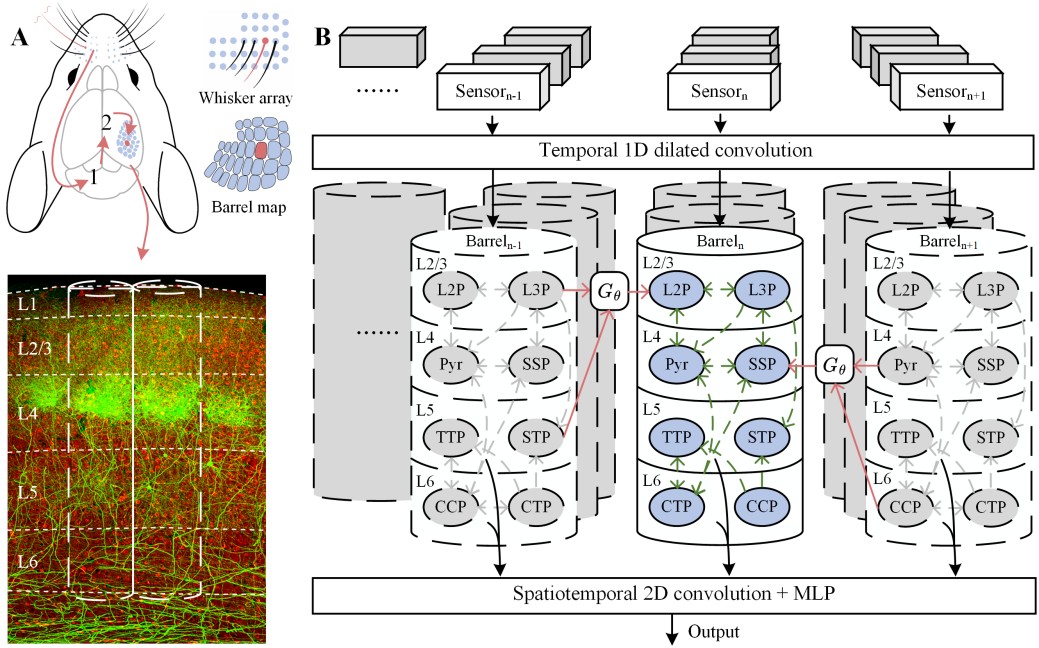

Figure 1: Outline of our barrel cortex-constrained localist expert model. **(A)** Neural pathway of "whisker-to-barrel" processing in rodent. Top: whisker signals are transmitted via (1) brainstem neurons and relayed through (2) thalamic neurons to the barrel cortex. Each processing stage (brainstem, thalamus, and barrel cortex) contains somatotopically organized units corresponding to individual whiskers, with the barrel cortex exhibiting hierarchical cortical columns (modified from [37]). Bottom: microscopic image of the barrel cortex, clearly showing horizontally arranged columns and vertically stratified layers. **(B)** Computational architecture of our multi-barrel model. Each barrel column incorporates 37 synaptic connections across eight neuronal subtypes, derived from established neuroscience studies, strictly adhering to the one-to-one "whisker-barrel" mapping by processing only its assigned tactile sensor's data. Neuronal subtype labels, from layer 2 (L2) to L6, correspond to their neuroscientifically defined nomenclature (e.g., "SSP" denotes L4 spiny stellate pyramidal neurons [9, 43]). Neuronal subtype details in supplementary materials. Adjacent barrels integrate inter-barrel currents through a dynamic gating network. The model employs temporally dilated 1D convolutions [17] to simulate brainstem and thalamic preprocessing without inter-sensor leakage, utilizes 2D convolutions to integrate L5/6 neuronal states across barrels mimicking cortical-subcortical projections, and applies a multilayer perceptron (MLP) for final predictions.

By contrast, contemporary artificial neural networks generally do not impose local specialization. Architectures such as convolutional neural networks (CNNs) [26, 46] and Transformers [49, 8] are predominantly global in their computational flow, lacking explicit modular substructures tied to specific sensory channels. The recent success of Mixture-of-Experts (MoE) models [11, 34, 7, 29] has revived interest in modular computation in artificial intelligence. These models use a gating mechanism to activate a sparse subset of sub-networks ("experts") per input, improving both efficiency and scalability. Nevertheless, the routing strategy in artificial MoEs fundamentally differs from that of the brain. Artificial MoEs focus on task allocation—not enforced inter-expert collaboration and communication. In contrast, the brain utilizes topographic and localist routing, where each expert (e.g., cortical column) processes only its corresponding sensory signal and communicates primarily with spatially adjacent modules. This contrast raises a key question: can we construct a MoEs-like model based on the biologically grounded routing constraints observed in the brain?

In this work, we propose a barrel cortex-inspired modular expert network for sensorimotor learning. The architecture consists of 39 interacting modules, each corresponding to a tactile sensor (whisker), faithfully replicating the somatotopic organization of the rodent barrel cortex. Each module processes its designated sensory stream using an internal structure composed of multiple biologically inspired neuron subtypes, mirroring cortical L2 through L6 (Fig. 1B). All modules share a common architecture

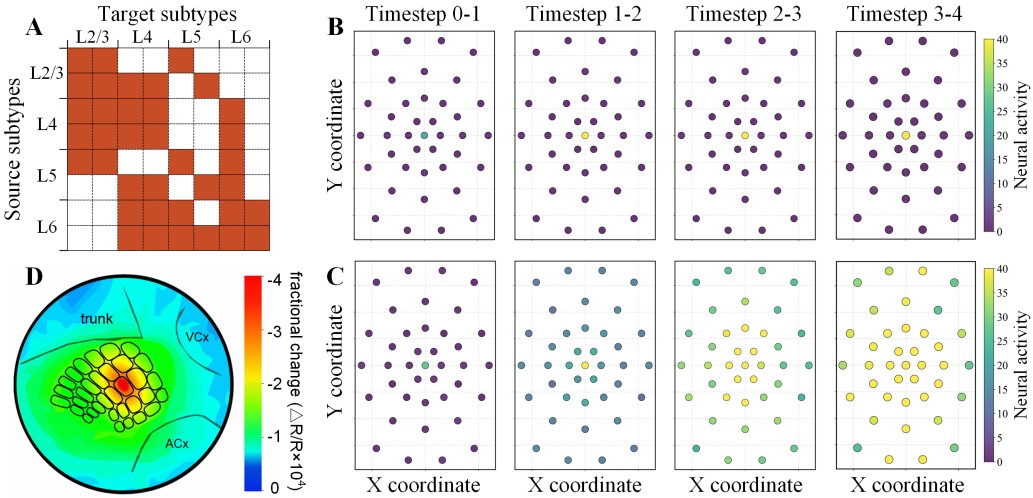

Figure 2: Horizontal spread of neural activity in the multi-barrel model. **(A)** Schematic of vertical connectivity generalized from published neuroscience studies. Red lines indicate synaptic connections between distinct neuronal subtypes. **(B)** Temporal evolution of barrel states ($\frac{spikes}{\Delta T}$) in the initial model without inter-barrel currents, where only the central barrel receives sustained current input from its corresponding sensor. Barrel layout reflects the spatial arrangement of the sensor array. **(C)** Same as in (B), but with inter-barrel currents enabled in the model. **(D)** Experimentally observed activation spread across barrels in real optogenetic recordings (adapted from [23]).

and parameter set, reflecting the repeated structure observed across cortical columns, and enabling substantial reductions in parameter count. To enable local integration while preserving spatial structure, we introduce a topographic gating mechanism: each module exchanges information only with its immediate neighbors, reproducing the localized lateral connectivity found in the barrel cortex. This results in sparse, spatially constrained expert activation, in sharp contrast to the global routing strategies used in classical MoE frameworks. Our key contributions are summarized as follows:

- **Biologically constrained modular architecture:** We introduce a modular neural network whose submodules (barrels) replicate the canonical microcircuitry and sensor-mapping of the rodent barrel cortex. Each module integrates multiple neuron subtypes and strictly processes signals from its designated sensor, enabling biologically faithful modular computation.

- **Localist expert routing:** We implement a spatially grounded gating mechanism that enables communication between adjacent modules only, contrasting with current MoEs that neglect inter-expert routing. Experimental results demonstrate that localist connection suppress activity correlations between modules while reducing functional connection distances.

- **Empirical validation on tactile tasks:** Our biologically inspired model achieves state-of-the-art performance on challenging 3D tactile classification datasets [16]. The model reduces parameter usage by 97% through shared columnar weights, while improving training stability. These results demonstrate the viability of embedding strong biological priors into scalable machine learning systems.

Our work positions the barrel cortex as a neurobiologically constrained realization of an expert system, providing a conceptual and empirical bridge between cortical computation and modular artificial intelligence. By integrating neuroscience-inspired modularity with sparse local interactions, we offer a new perspective on the design of efficient, interpretable, and high-performing artificial systems. We argue that principles such as topographic routing, localist specialization, and structural sparsity—long favored by evolution—can inform the development of next-generation machine learning architectures with more brain-like sensorimotor intelligence.

## 2 Related work

**Barrel cortex model:** Due to its highly specialized structure and unique tactile functions, the rodent barrel cortex has become a preferred model system for neuroscientists studying sensory perception. Through techniques such as immunohistochemistry, microscopic imaging, and neuronal reconstruction, considerable knowledge has been accumulated on its cellular distribution patterns [36, 48, 42] and vertical [44, 12, 31]/horizontal [37, 23, 13, 3] circuit organization. Based on these anatomical data, computational neuroscientists have developed local circuit models [2, 22, 25] and detailed dendritic models [28] to investigate network dynamics. However, these brain simulation models lack learning capabilities, and few incorporate multi-barrel column architectures. Recently, a trainable superficial layer model that maintains biological plausibility while demonstrating tactile processing functions was proposed [59], though it remains limited to single-barrel representation without incorporating "whisker-barrel" topographic mapping.

**Columnar machine learning models:** The uniform yet distinctly layered structure of the neocortex, widely reported in neuroscience, has also drawn attention in machine learning. Several studies [19, 18, 40] have mimicked the columnar organization of the neocortex by designing analogous basic units that exhibit hierarchical feature abstraction and local-to-global integration. However, their information pathways are governed by simplistic rules, resulting in limited performance. Recent columnar models [55, 24] have prioritized hardware-friendly physical implementations to improve computational efficiency. However, these models overlook the fact that cortical column functionality is based on complex and diverse neural circuits [21], and still fall short of matching the performance of contemporary machine learning systems. Meanwhile, loosely modular architectures like MoE [11, 7, 34, 29] in large-scale models are gaining prominence. A promising future direction lies in the simulation of the collaborative paradigm of neocortical regions to construct versatile expert systems capable of multifunctional integration.

## 3 Methodology

### 3.1 Biologically constrained columnar modules

To enable scalable yet biologically plausible modular replication, we model barrel columns at the neural pathway-level. While prior single column models [4, 6, 33] achieved one-to-one biological fidelity through massive neuron counts (requiring cluster computing), our architecture strategically incorporates 8 excitatory neuron subtypes and 37 documented projection pathways from barrel cortex studies [37, 44, 12, 14, 31, 39, 27, 58], including projections from Layer 4 (L4) to L2/3 neurons and feedback connections from L5/6 to L4 neurons, among others (Fig. 1B). To enhance scalability, we: (1) represent inhibitory effects through negative weights rather than explicit interneurons, and (2) implement each subtype with 32 cells (256 neurons/barrel), a configuration that maintains biological plausibility while enabling efficient scaling to multiple barrels. Detailed information on the eight summarized neuronal subtypes is provided in the supplementary materials.

Neuronal dynamics follow an adaptive Leaky Integrate-and-Fire (aLIF) model [54] with background currents. The membrane potential $V$ update equation is defined as follows:

$$V^{(t)} = e^{-\frac{1}{\tau_m}} \cdot V^{(t-1)} + (1 - e^{-\frac{1}{\tau_m}}) \cdot R_m \cdot I^{(t)} - S^{(t-1)} \cdot \theta^{(t-1)}$$

$$I^{(t)} = I_e^{(t)} + I_r^{(t)} + I_{agg}^{(t)} + \epsilon^{(t)}, \quad \epsilon^{(t)} \sim \mathcal{N}(0, 1)$$

$$S^{(t)} = \Theta\left(V^{(t)} - \theta^{(t)}\right), \quad \Theta(x) = \begin{cases} 1 & x > 0 \\ 0 & \text{otherwise} \end{cases} \tag{1}$$

, where $R_m$ and $\tau_m$ represent the membrane resistance and time constant, respectively. The input currents $I$ are categorized into four distinct components: $I_e$ denotes external input currents, $I_r$ represents intra-barrel synaptic currents, $I_{agg}$ corresponds to aggregated inter-barrel currents from neighboring columns, and $\epsilon$ denotes the background noise sampled from a standard normal distribution. When the membrane potential $V$ exceeds the firing threshold $\theta$, the neuron emits a spike $S$. The neuronal firing threshold undergoes an adaptive elevation through spike-triggered accumulation, governed by the following dynamics:

$$\theta^{(t)} = \theta_{init} + \beta \cdot \eta^{(t)}$$

$$\eta^{(t)} = e^{-\frac{1}{\tau_{adp}}} \cdot \eta^{(t-1)} + (1 - e^{-\frac{1}{\tau_{adp}}}) \cdot S^{(t-1)} \tag{2}$$

Table 1: Comparison across methods on two tactile datasets [16]. *Independent* and *Shared* denote our model's 39-barrel configurations with independent and shared parameters, respectively. *Single* refers to a single barrel model with equivalent neuron size (39×256).

| Method | EvTouch-Objects (%) | EvTouch-Containers (%) |
|---|---|---|
| TactileSGNet [16] | 89.44 | 64.17 |
| Grid-based CNN [16] | 88.40 | 60.17 |
| GCN [16] | 85.14 | 58.83 |
| Method in [51] | 90.28 | - |
| SnnTdlc [56] | 91.04 | 67.33 |
| AM-SGCN [52] | 91.32 | - |
| GGT-SNN [53] | 92.36 | 75.00 |
| Single barrel model | 88.89 | 70.00 |
| Independent multi-barrel model | 92.36 | 85.00 |
| Shared multi-barrel model | **94.44** | **86.67** |

, where $\theta_{init}$ and $\tau_{adp}$ denote the resting firing threshold and adaptation time constant, respectively, with $\beta$ being a constant parameter set to 1.8. The recurrent current $I_r$ received by a postsynaptic neuronal population from other upstream populations within the home column can be computed as:

$$I_r^{(t)} = \sum_{j \in P} W_j \cdot S_j^{(t)}, \quad W_j \in \mathbb{R}^{32 \times 32} \tag{3}$$

, where $W_j$ and $S_j^{(t)}$ denote the connection weight matrix and the spike vector of the $j$-th upstream population in the presynaptic set $P$, respectively. The computation processes for $I_{agg}$ and $I_e$ in Eq. 1 are elaborated in the following two sections.

### 3.2 Localist inter-barrel routing

Beyond vertical intra-barrel pathways, the barrel cortex employs horizontal inter-barrel connections for sensory integration. Neuroanatomical evidence indicates that lateral connections follow an adjacency-priority principle [32, 12], with inter-barrel signaling predominantly originating from spatially adjacent columns and minimal contributions from distant ones. We implemented this biologically observed connectivity through K-nearest neighbor (KNN) spatial mapping coupled with sparse gating mechanisms.

For a 32-neuron subtype in our model, input signals from neighboring barrels follow these governing equations. First, the most relevant neighboring subtypes indices $\mathcal{T}$ are dynamically selected based on gating weights:

$$\mathcal{T}^{(t)} = TopK(G^{(t)}, \Gamma), \Gamma = \lfloor K \cdot 8 \cdot \gamma \rfloor$$
$$G^{(t)} = MLP_\phi(I_e^{(t)}) \in \mathbb{R}^{K \times 8} \tag{4}$$

, where $K$ denotes the number of spatially adjacent barrels (nearest neighbors). $\gamma$ represents the fraction of total $(K \cdot 8)$ neuronal subtypes to be routed, giving $\Gamma$ selected subtypes. $I_e^{(t)}$ and $\phi$ correspond to external input currents and trainable parameters, respectively. $TopK(\cdot)$ returns a binary mask $\in \{0,1\}^{K \times 8}$ with exactly $\Gamma$ nonzeros. The spatial distance between barrels is computed based on the sensor coordinates provided in the tactile dataset [16]. $K$ and $\gamma$ were assigned values of 4 and 0.2, respectively.

Then, the final aggregated neighborhood current $I_{agg}^{(t)}$ is computed by applying the selected subtypes indices from $\mathcal{T}^{(t)}$ (mask) to the spiking states $S_g^{(t)}$, flattening the results to a vector in $\mathbb{R}^{\Gamma \cdot 32}$, and performing a trainable linear transformation:

$$I_{agg}^{(t)} = W_{agg} \cdot Flatten(\mathcal{T}^{(t)} \odot S_g^{(t)}) \tag{5}$$

, where $W_{agg} \in \mathbb{R}^{32 \times (\Gamma \cdot 32)}$ denotes the trainable weight matrix mapping the flattened neighborhood activity vector to the 32 neurons of the target barrel, and $S_g^{(t)} = [s_1^{(t)}, s_2^{(t)}, ..., s_{K \times 8}^{(t)}] \in \mathbb{R}^{(K \cdot 8) \times 32}$

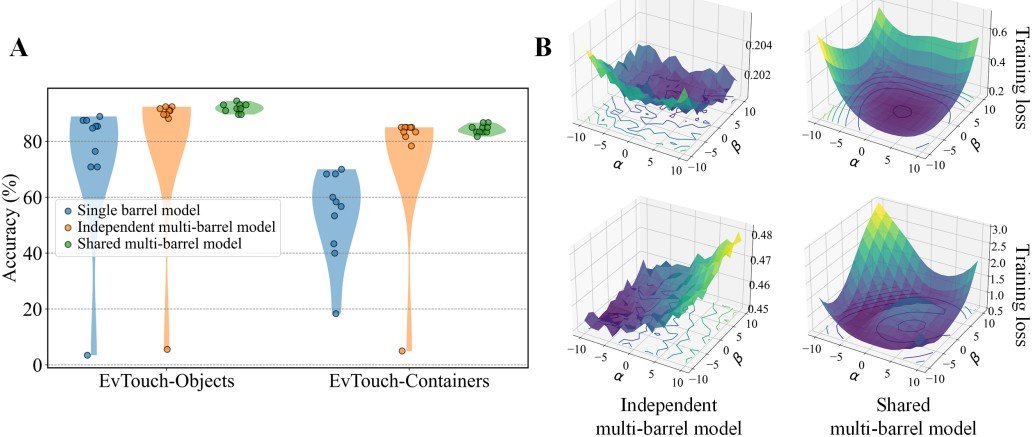

Figure 3: Performance comparison of three model variants. **(A)** Scores from 10 repeated random training runs for each variant. **(B)** Loss landscapes of independent-parameter and shared-parameter models. Perturbations were applied along two orthogonal directions to trained parameters: $\theta^{'} = \theta + \alpha\eta_1 + \beta\eta_2$, where $\theta$ and $\theta^{'}$ denote the original and perturbed parameters. $\alpha$ and $\beta$ are perturbation magnitudes. Top to bottom: results on the EvTouch-Objects and EvTouch-Containers datasets.

represents the tensor composed of spiking states from all neuronal subtypes in $K$ neighboring barrels at timestep $t$.

Fig. 2B-C illustrates the temporal state evolution in our multi-barrel model under optogenetic-like stimulation [1, 15], achieved through constant current activation of the central sensorimotor barrel. The results reveal that incorporating inter-barrel current coupling induces propagated activation from the stimulated barrel to neighboring regions, progressively diffusing across the entire array. This spatiotemporal propagation pattern aligns with cortical barrel dynamics reported in empirical optogenetic studies of the "whisker-barrel" somatosensory system [23, 13, 3] .

### 3.3 Empirical validation on tactile datasets

We aim to both develop a barrel cortex-inspired architecture and validate the "whisker-barrel" system as a localist expert processor. Below, we describe the public datasets used and our differentiable readin-readout implementation, which bridges biological computation with machine learning optimization through effective gradient propagation.

Two tactile datasets similar to whisker systems, EvTouch-Objects and EvTouch-Containers [16], were employed to benchmark our model against artificial neural networks. These datasets feature temporal signals recorded from independent NeuTouch sensor arrays [47] during interactions with diverse 3D objects, requiring the model to predict object categories based on dynamic tactile inputs. Each data sample has a shape of $[39, 2, T]$, representing two channel signals recorded from 39 sensors over T timesteps. Appendix A.1 provides complete tactile dataset specifications.

We treat each sensor as a rodent whisker (Fig. 1B), where brainstem-thalamic signal preprocessing [41, 5] is simulated via 2 one-dimensional dilated convolutional layers [17]. A linear layer then serves as the thalamic signal relay. The external current $I_e$ received by a neuronal subtype in barrel cortex is computed as:

$$I_e^{(t)} = W_{th}^T \cdot (W_{c2} \circledast \sigma(W_{c1} \circledast X^{(t)})), \quad W_{th} \in \mathbb{R}^{64 \times 32} \tag{6}$$

, where $X$ and $\circledast$ denote the input data and dilated convolution operation, respectively. $W_{c1}$ and $W_{c2}$ represent the first- and second-layer convolution kernels. $\sigma$ is the activation function. The convolution operates exclusively along the temporal dimension, preventing inter-sensor information leakage. The output tensor dimension becomes $[39, 64, T^{'}]$, where $T^{'} = \lfloor \frac{T - d(k-1) - 1}{s} \rfloor + 1$ with $d$, $k$ and $s$ being the dilation rate, kernel size and stride, respectively.

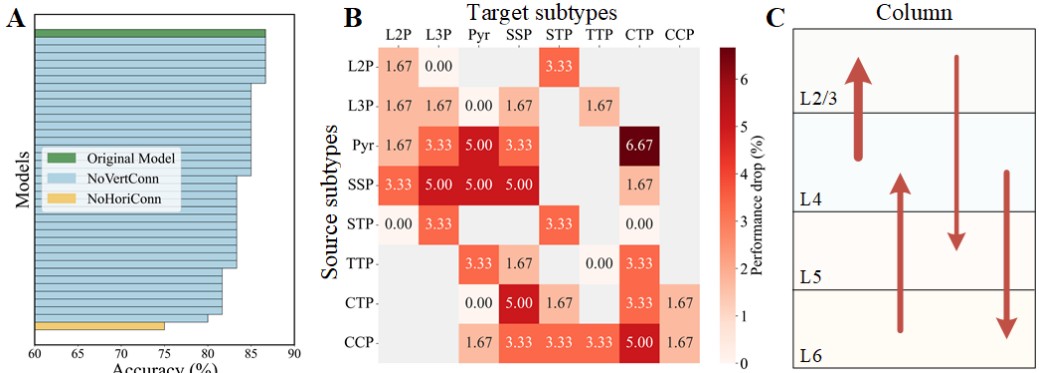

Figure 4: Ablation analysis of synaptic connectivity on EvTouch-Containers. **(A)** Model performance after isolated removal of vertical connections and horizontal inter-barrel currents, ranked by descending accuracy. **(B)** Quantitative importance of 37 intra-barrel connections, measured by performance degradation magnitude. **(C)** Four highest-impact signaling pathways identified from (B).

Given the dominant role of L5/6 neurons in driving subcortical projections within barrel cortex [12, 44], we read-out the state of our model's L5/6 neuronal populations through: (1) spatiotemporal integration via a two-dimensional convolution layer, followed by (2) final classification prediction through a multilayer perceptron (MLP). The formula is expressed as follows:

$$\hat{y} = Softmax(W_{mlp} \cdot Flatten(W_{c3} \ominus S_{L5/6})), \quad S_{L5/6} \in \mathbb{R}^{T' \times 39 \times (32 \cdot 4)} \tag{7}$$

, where $\ominus$ denotes the two-dimensional convolution operation. $W_{c3}$ and $W_{mlp}$ represent the convolution kernel and MLP parameters, respectively. $S_{L5/6}$ denotes the spiking states of four neuronal ensembles (32 neurons each) in L5/6 across 39 barrels over $T'$ timesteps. The model employs standard cross-entropy loss, with spiking neuron gradients computed via the Gaussian surrogate function: $\mathcal{N}(V^{(t)}|\theta^{(t)}, \sigma^2)$, where $\sigma$ is set to 0.5.

Given that our multi-barrel model is extended from a single barrel as introduced in Sec. 3.1, a natural question arises: whether these barrels **share training parameters** or maintain their own independent parameters. Excluding the shared readin and readout pathways, in our 39-barrel configuration, the parameter count of the column modules with shared parameters is reduced by approximately 97% compared to those with independent parameters. In practice, their parameter counts are 59,104 and 2,305,056, respectively, a reduction of two orders of magnitude. This significant decrease in parameters is crucial not only for computational efficiency, but also serves as a form of regularization. In the experimental section, we investigate the tactile task performance of both model variants.

## 4 Experimental results

### 4.1 Implementation details

Our multi-barrel model was trained for 200 epochs on both EvTouch-Objects and EvTouch-Containers datasets [16] using an AdamW optimizer [30] with 0.1 weight decay. The initial learning rate of 0.0008 decayed by a factor of 0.8 every 10 epochs. The loss function employed solely the standard cross-entropy criterion. All experiments were conducted on an 80GB NVIDIA A100 GPU.

### 4.2 Performance comparison with baseline methods

First, we evaluated our multi-barrel model against reported baselines on both EvTouch-Objects and EvTouch-Containers datasets, achieving state-of-the-art (SOTA) classification accuracy (Tab. 1). Our model outperforms the previous best-performing method GGT-SNN [53], achieving accuracy improvements of 2.1% on EvTouch-Objects and 11.7% on EvTouch-Containers. While prior work employed graph neural networks [16, 52, 53] to process sensor topology, our biologically constrained architecture attains comparable performance through simple "whisker-to-barrel" mapping. Notably,

collapsing the columnar organization into a single barrel reduces accuracy by 5.6% (EvTouch-Objects) and 16.7% (EvTouch-Containers), demonstrating the functional importance of modular architecture.

Furthermore, we compared two model variants from Sec. 3.3: parameter-shared (39 barrels) versus independent-parameter configurations. The shared-parameter variant showed superior performance (2.1% higher on EvTouch-Objects, 1.7% on Containers) with greater training stability, whereas independent-parameter model occasionally failed to train (Fig. 3A). Loss landscape analysis reveals this dichotomy: shared parameters create smooth basins enabling robust convergence, whereas independent parameters yield rugged terrain with local fluctuations (Fig. 3B). This likely occurs because independent parameters per barrel generate conflicting gradient updates during training, thereby complicating optimization. In contrast, shared parameters naturally enforce cross-barrel regularization, effectively mitigating stochastic gradient variations. These results validate the functional advantages of biologically uniform organization and modular replication in artificial expert systems.

## 4.3 Ablation analysis of synaptic connectivity

Then, we systematically assess synaptic importance in our multi-barrel model through ablation experiments, isolating and blocking each connection while quantifying performance degradation. Taking the EvTouch-Containers dataset as an example, inter-barrel aggregated currents show the greatest impact, reducing classification accuracy by approximately 11.7% (Fig. 4A), while some vertical connections exhibit minimal impact.

Fig. 4B demonstrates laminar-specific effects through directional analysis of 37 intra-barrel vertical projections. The four L4 to L2/3 connections collectively contributed 13.3% to performance, whereas two reciprocal L2/3 to L4 connections showed weaker impacts but stronger L5 modulation. We identified four dominant pathways: L4-L2/3, L4-L6, L2/3-L5, and L6-L4 (Fig. 4C). These pathways align with established neuroanatomical principles: the L4-L2/3-L5 microcircuit represents a canonical barrel cortex circuit [37, 31], and L6 cortico-thalamic pyramidal (CTP) cells receive dendritic inputs from L4 pyramidal neurons [39, 44] and project reciprocally to L4 boundaries [27, 58]. This alignment may suggest that cortical microcircuits represent an evolutionarily optimized architecture.

Besides, deactivating specific synaptic connections (e.g., STP to L2P) does not compromise model performance, suggesting the existence of redundant pathways. This functional resilience likely results from substitution by connections with similar directional properties. Consequently, more precise characterization is required to delineate the distinct contributions of specific neuronal subtypes.

## 4.4 Analysis of localist routing patterns

Next, we examined the localist routing behavior of our multi-barrel model. In contrast to the decentralized, globally routed structure of conventional MoE systems, each expert (i.e., barrel column) in our system communicates only with its immediate neighbors. As illustrated in Fig. 5A, the KNN gating constraints introduced in Sec. 3.2 yields predominantly short-range horizontal connections in both the initial and the trained models; long-range projections spanning multiple barrels are entirely absent.

To quantify the functional impact of these horizontal connections, we analyzed the correlations of barrel-column activity on the testsets, considering both full-trial and sliding-window time scales (Fig. 5B). Introducing inter-barrel currents decreased global mean correlations by 0.07 on EvTouch-Objects and 0.01 on EvTouch-Containers; windowed (local) correlations fell by 0.03 and 0.01, respectively (Fig. 5E). Reduced synchrony suggests that horizontal currents promote richer, more specialized activity patterns across columns, thereby improving the discriminability of complex inputs. This effect echoes the lateral-inhibition mechanism reported in the biological barrel cortex, where activated columns transiently suppress their neighbours to sharpen sensory contrast [38, 35, 10].

The spatial signature of these effects is equally striking. Inter-barrel currents shortened the average distance between strongly correlated column pairs by 0.06 grid units on EvTouch-Objects and 0.17 on EvTouch-Containers; for the top-5 most-correlated pairs, the reductions were even larger—1.8 and 3.48 units, respectively (Fig. 5E). As depicted in Fig. 5C-D, the top-correlation pairs overwhelmingly involved neighbouring barrels once horizontal gating was enabled. Together, these findings show that localist inter-barrel currents simultaneously strengthen short-range functional connectivity and amplify activity differentiation across columns. The results are consistent with long-standing neuro-

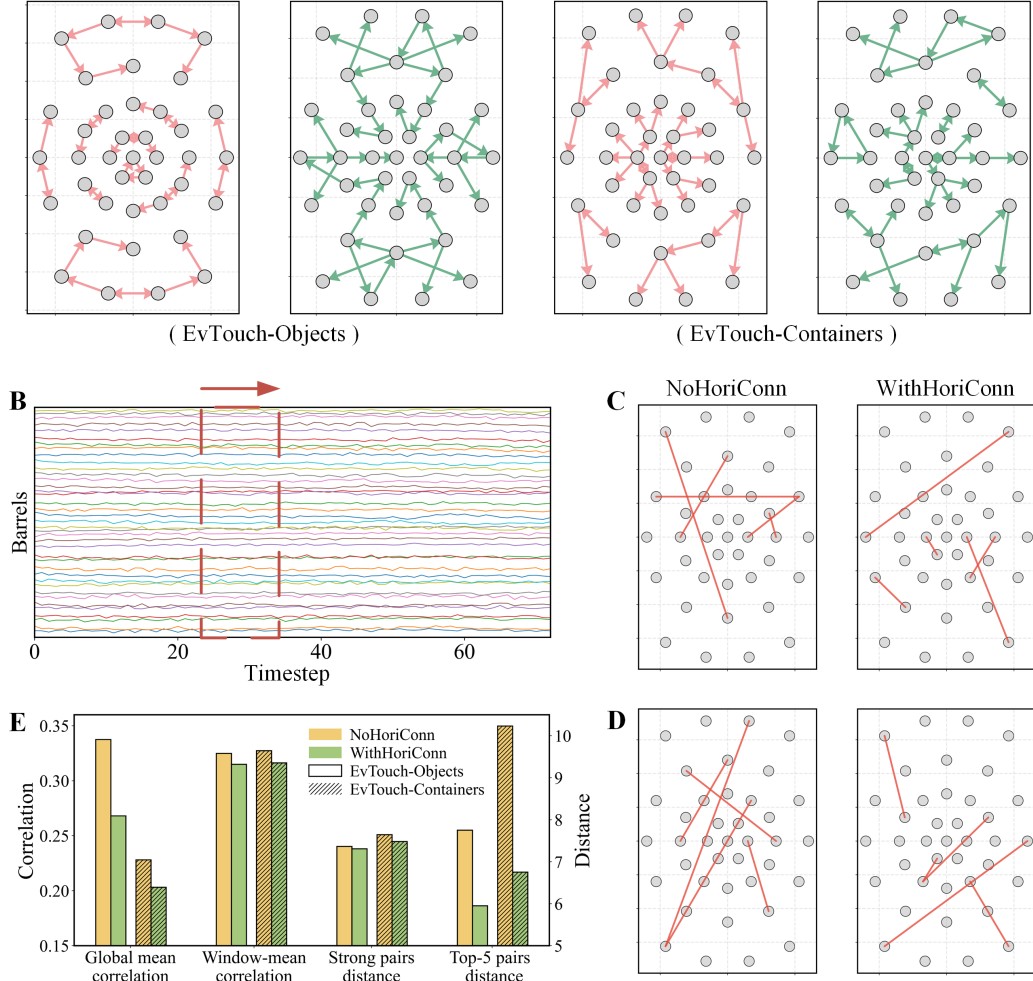

Figure 5: Statistic of inter-barrel correlations. **(A)** Barrel-wise localist routing. Arrows indicate the most frequently selected source neighbor barrels, as determined by cumulative counts across samples and timesteps. **(B)** Neural activity of 39 barrels during a single trial. A sliding window (length = 10, stride = 2) was applied to compute pairwise activity correlations. The three most strongly correlated barrel pairs per window were selected as high-correlation pairs. **(C)** Top-5 most frequent strongly correlated barrel pairs on EvTouch-Objects dataset. From left to right are models with blocked and intact horizontal inter-barrel currents. **(D)** Same as (C), but for EvTouch-Containers dataset. **(E)** Statistical metrics between blocked and intact models. From left to right are whole-time mean barrel correlations, windowed mean correlations, mean distances of strongly correlated barrel pairs, and mean distances of Top-5 pairs.

physiological principles [32, 12] and may inform the design of future MoE systems with enforced inter-expert communication to reduce information redundancy [50, 57].

## 5 Conclusion

In this work, we focus on the rodent barrel cortex as a localist expert system in the brain, developing a multi-barrel model that strictly adheres to the one-to-one "whisker-barrel" somatotopic mapping. Our architecture faithfully replicates the laminar and columnar organization of barrel cortex, achieving state-of-the-art performance on two tactile datasets while successfully balancing biological plausibility and behavioral performance. Experimental results demonstrate that the cortex's uniform modular

architecture facilitates parameter sharing to enhance training stability, while localist routing reduces functional connection distances and suppresses inter-barrel activity correlations. Inter-barrel currents may sharpen perception and reduce redundancy. This work reveals the potential of brain's native expert systems to inspire next-generation machine learning architectures.

**Limitations and future work:** On one hand, the redundant synaptic connections observed in Fig. 4B necessitate more refined neuronal subtype differentiation in modeling, which requires deeper anatomical knowledge integration. On the other hand, our multi-barrel model remains distinct from standard MoE architectures and has not yet been validated on mainstream machine learning benchmarks. Developing brain-inspired MoE architectures that emulate cortical coordination mechanisms presents an exciting research direction. Another promising future direction involves embedding our biologically constrained barrel cortex model into realistic real-time interactive environments. This framework would leverage biologically constrained models to explore performance in complex sensory tasks (e.g., navigation), thereby significantly advancing both our understanding of brain operating principles and potential applications.

## Acknowledgments

This work was financially supported by the STI 2030-Major Projects (2021ZD0201002), and National Natural Science Foundation of China grants (T2122015, 32471149).

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

# A    Technical Appendices and Supplementary Material

## A.1    Tactile task introduction

This section details our tactile perception task. Both employed datasets, EvTouch-Objects and EvTouch-Containers [16], were acquired through an identical protocol: the NeuTouch [47] sensor array engaged in several-second tactile interactions with 3D objects, followed by object category prediction based on the recorded sensor signals (Fig. 6A). The NeuTouch system integrates 39 uniformly configured sensor units, each generating two-channel spiking signals, with their spatial arrangement depicted in Fig. 6B. The datasets were categorized into EvTouch-Objects (720 samples, 36 categories) and EvTouch-Containers (300 samples, 20 categories) based on object taxonomy, with both datasets split into training and test sets at an 8:2 ratio for standardized evaluation.

As described in Sec. 3.3, we modeled each of the 39 tactile sensors as a rodent whisker, with separate barrel modules processing each sensor's signals independently. Each raw sample in the tactile dataset has a shape of [39, 2, T], representing two-channel signals from 39 sensors over T timesteps, where T equals 250 for EvTouch-Objects and 325 for EvTouch-Containers. We first applied two 1D dilated convolutions along the temporal dimension to simulate brainstem and thalamic preprocessing of whisker signals [41, 5], including delay and integration effects [17]. Dilated convolutions were exclusively applied along the time axis to prevent cross-whisker signal leakage (Fig. 6C). The processed timesteps computed as $T^{'} = \lfloor \frac{T-d(k-1)-1}{s} \rfloor + 1$, where $d$, $k$ and $s$ represent the dilation rate, kernel size, and stride respectively. Then, the preprocessed whisker signals were independently propagated to their corresponding barrel columns. Finally, a standard 2D convolution operation integrated the barrel-wise L5/6 neuronal states across the entire array, followed by an MLP to generate the final prediction output.

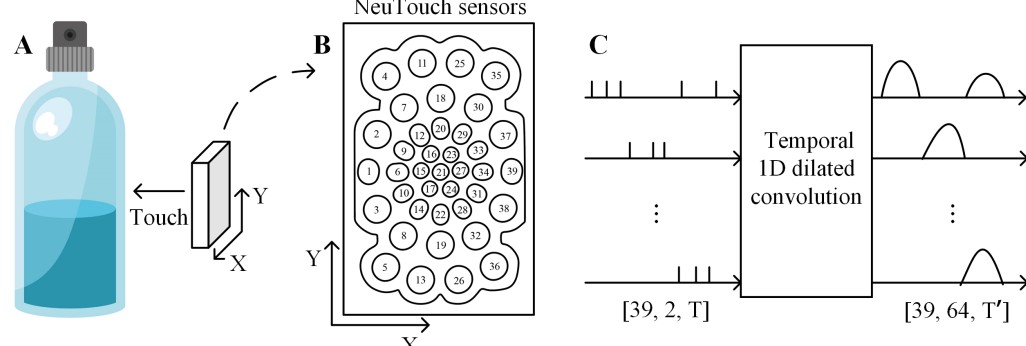

Figure 6: Tactile perception task overview. **(A)** Object categorization via several-second tactile scanning using NeuTouch sensor array (datasets: EvTouch-Objects/Containers) [16]. **(B)** NeuTouch's 39-sensor spatial configuration (adapted from [47]). **(C)** Each sensor modeled as a whisker, with 1D dilated convolutions simulating brainstem-thalamic delay/integration [17, 41, 5].

