# OpenReview forum: "Localist Topographic Expert Routing: A Barrel Cortex-Inspired Modular Network for Sensorimotor Processing"
_NeurIPS.cc/2025/Conference — NeurIPS 2025 poster_

### Official Review · Reviewer_s3by · 2025-07-02

**Clarity:** 4
**Significance:** 3
**Originality:** 3
**Rating:** 5
**Confidence:** 4

**Summary:**

This paper proposes a barrel cortex-inspired modular network with localist topographic expert routing for sensorimotor processing. The architecture strictly adheres to biological constraints (one-to-one "whisker-barrel" mapping, vertical/horizontal microcircuits) and achieves SOTA performance on 3D tactile tasks with 97% parameter reduction. While biologically innovative and technically sound, broader validation and theoretical grounding are needed for stronger impact.

**Questions:**

1. Can the architecture maintain its efficiency/performance advantages on non-tactile tasks?
example:
Test on two new task types:
(a) Sensorimotor: Audio event classification (e.g., DCASE datasets) using spectrogram "topography"
(b) Non-sensorimotor: Image classification (e.g., CIFAR-10) with spatial patch → expert mapping

2. Do biological constraints provide tangible benefits over standard MoEs?
example:
Add 3 baselines with matched params:
(a) Switch Transformer (global routing)
(b) Top-k MoE with fixed neighbors (non-adaptive)
(c) Dense model (all experts active)

3. Which constraints are essential vs. redundant?
example:
Run ablations:
(a) Remove topographic routing → random neighbor connections
(b) Remove vertical microcircuits → replace with MLP
(c) Remove parameter sharing → independent weights
Measure performance drop on EvTouch-Containers

4. Does performance degrade with scaled sensor arrays?
example:
Simulate larger arrays (78/156 sensors) via:
(a) Sensor duplication + noise
(b) Zero-padding unused inputs

**Ethical Concerns:**

["NO or VERY MINOR ethics concerns only"]

**Final Justification:**

The study demonstrates impressive innovation; however, it would benefit from additional exploration of the optimal trade-offs between simplicity and biological fidelity. In summary, while the work is acceptable and merits publication, it exhibits certain limitations that prevent it from achieving spotlight-level recognition.

**Limitations:**

The authors should more systematically explore the conditions under which their method works effectively and where it may fail. This enhances understanding and application of the model by other researchers and guides future research. For instance, experimenting with different input data types can analyze the model's robustness and adaptability. Also, studying the model's performance under extreme conditions or with abnormal inputs can reveal its potential limitations.

Regarding the computational overhead from biological constraints, the authors should conduct an in - depth analysis of the increased computational overhead from the biological constraints in the model design. This includes evaluating their impact on computational resource requirements and model efficiency during training and inference. To boost the model's practicality, methods to optimize computational processes and reduce resource consumption while maintaining biological plausibility should be explored.

**Quality:**

3

**Strengths And Weaknesses:**

Strengths
1. Novel Bio-Inspired Architecture:
Faithfully translates rodent barrel cortex principles (somatotopic organization, laminar microcircuits) into a modular MoE-like system (§3.1–3.2).

2. Localist routing via KNN gating effectively suppresses inter-module correlations, mirroring cortical lateral inhibition (§4.3, Fig. 5).

3. 97% parameter reduction via columnar weight sharing (§3.3) while improving accuracy over baselines (+2.1–11.7% on EvTouch datasets, Table 1).

4. Loss landscape analysis confirms enhanced training stability (§4.2, Fig. 3B).

5. Spatiotemporal activation patterns align with optogenetic studies (§3.2, Fig. 2C-D).

6. Ablation studies validate neuroanatomical principles (L4→L2/3 dominance, Fig. 4C).

Weaknesses
1. Limited Generalizability (Critical):
Evaluated only on tactile tasks (EvTouch). Unclear if the architecture generalizes to other modalities (e.g., vision/audio) or non-sensorimotor domains.

2. Insufficient Comparison with MoE Baselines:
Lacks direct comparison to standard MoEs (e.g., Switch Transformer) to quantify benefits of biological constraints (routing efficiency, expert diversity).

3. Methodological Clarity Issues:
Undefined tensor dimensions (e.g., Eqs in §3.2) and incomplete figure annotations (Fig. 2 temporal evolution).

4. Inadequate explanation for why biological constraints improve performance beyond empirical results.

---

> ### Author Rebuttal · Authors · 2025-07-29
>
> Thank you for your careful review and valuable comments. We will improve the relevant descriptions in the revised manuscript (e.g., the equations in Sec. 3.2 and the caption for Fig. 2). Below, we would like to respond to your questions point by point.
>
> > Q1: Can the architecture maintain its efficiency/performance advantages on non-tactile tasks? example: (a) Audio event classification... (b) Image classification...
>
> **A1:** In the early stages of exploring the single barrel column structure, we conducted preliminary tests on visual datasets (MNIST and Fashion-MNIST) and an auditory dataset (SHD). However, key challenges arose when extending the model to a multi-barrel design: the central concept of the multi-barrel model relies on the one-to-one "whisker-barrel" mapping, which is absent in visual and auditory datasets. While the alternatives you proposed, such as using *"spatial patches"* or *"spectrograms"*, **are technically feasible, they raise the following conceptual concerns:**
> 1. Do visual and auditory signals possess a natural modular topological structure, similar to tactile signals?
> 2. Are visual and auditory signals primarily reliant on local dependencies? Long-range dependencies are also common.
>
> These questions suggest that the proposed localist routing mechanism may not immediately align with visual and auditory datasets. In fact, different cortical regions have evolved distinct structures to handle tasks across different modalities. For example, the barrel cortex is specialized for processing tactile signals, whereas the auditory cortex has no reported discrete modular structures resembling barrels.
>
> But, we are not dismissing the value of extending the model to other domain tasks. On the contrary, such challenges inspire the abstraction of broader principles. For instance, while single samples in certain domains may lack an inherently modular structure, could their distribution in semantic space allow for natural clustering and measurable distances? Similarly, the localist routing mechanism in the barrel cortex intuitively bears conceptual similarity to the sliding window operation employed in convolutional neural networks. Overall, we agree the importance of investigating the model's generalizability to other modalities, but this lies somewhat beyond the scope of this paper's specific focus.
>
> ***
>
> > Q2: Do biological constraints provide tangible benefits over standard MoEs? Add 3 baselines (a) Switch Transformer...(b) Top-k MoE... (c) Dense model...
>
> **A2:** Following this valuable suggestion, **we added two MoE baseline models for comparison**. Considering that Switch Transformer is primarily designed for NLP tasks, we adapted standard simple MoE architectures — Top-2 MoE and Dense MoE — for tactile tasks. The gating network assigns signals from individual sensors to corresponding experts, with each expert designed as a 2-layer MLP containing 256 neurons in the hidden layer. Computation is performed independently across the time dimension. Multiple configurations with varying numbers of experts were tested, and each configuration was repeated three times. The best results are as follows:
>
> | Datasets |&#124;  | Top-2 MoE | |&#124; | Dense MoE | |&#124; Ours |
> |---|:---:|:---:|:---:|:---:|:---:|:---:|:---:|
> |          | 16 experts | 32 experts | 48 experts | 16 experts | 32 experts | 48 experts |      |
> | EvTouch-Objects | 88.89 | 90.28 | 90.97 | 90.28 | 89.58 | 88.89 | **94.44** |
> | EvTouch-Containers | 70 | 73.33 | 73.33 | 66.67 | 75 | 70 | **86.67** |
>
> As shown, both MoE baseline models perform markedly worse than our multi-barrel model across the tactile datasets. This disparity may arise from a lack of alignment between the simple MoEs and the characteristics of tactile data, as our model leverages a cortical-inspired one-to-one mapping. Additionally, the performance gap could be attributed to the limited representational capacity of the MLP units within the MoE experts.
>
> However, integrating localist routing into standard MoEs is challenging, as mapping the spatial topology of sensors to experts in a way that establishes meaningful neighborhood relationships is difficult. Achieving this would require adopting the one-to-one mapping from our multi-barrel model, but doing so would make it equivalent to the experiments in Q3(b), where the barrels in our model are replaced with MLPs. **We will detail these results in our response to Q3 (A3).**
>
> We believe you may be more interested in the performance of localist routing in large-scale MoEs on LLM benchmarks—a direction we are also eager to explore. However, this requires addressing critical questions: How should expert distances be defined in standard MoEs, perhaps in semantic space? And how should expert neighborhoods be structured, potentially through expert grouping?
>
> Our paper aims to highlight the key differences between AI expert systems (MoEs) and natural intelligence systems (e.g., the barrel cortex). Current MoE studies focus on task allocation, while the barrel cortex exhibits unique mechanisms of cooperation and competition. As future work, we aim to test brain-inspired MoE architectures on LLM benchmarks, though this would be another whole different work.
>
> ***
>
> > Q3: Which constraints are essential vs. redundant? example: Run ablations: (a)...(b)... (c)...
>
> **A3:** We agree that ablation experiments on biological constraints are important. Below are the detailed results for the three experimental configurations you suggested.
>
> **(a) Remove topographic routing → random neighbor connections**
> Following the suggestion, we kept the number of neighbors fixed at 4, replaced horizontal localist routing with random connections, and trained each model three times, reporting the best results in the table below. According to the results, this replacement caused performance drops of 4.2% and 3.3%, respectively. This is likely because object textures often exhibit spatial patterns, and localist routing preserves neighborhood information, enabling better integration of spatially correlated signals.  In contrast, random connections disrupt spatial organization, add noise, and weaken the model’s ability to capture meaningful relationships.  By leveraging the spatial coherence of tactile data, localist routing aligns better with the structured nature of sensory inputs, making it more effective.
>
> | Datasets | &#124; Random connection &#124; | Localist routing |
> |:---|:---:|:---:|
> |EvTouch-Objects|90.28|**94.44**|
> |EvTouch-Containers|83.33|**86.67**|
>
> **(b) Remove vertical microcircuits → replace with MLP**
> As suggested, we replaced each barrel column with MLPs of varying depths, each hidden layer containing 256 neurons. The model was trained three times, and the best results are recorded as follows:
>
> | Datasets | &#124; 2-layer MLP &#124; | 3-layer MLP &#124; | 4-layer MLP &#124; | Biological barrel |
> |:---|:---:|:---:|:---:|:---:|
> |EvTouch-Objects|89.58|88.89|88.89|**94.44**|
> |EvTouch-Containers|83.33|83.33|81.67|**86.67**|
>
> It is clear that replacing biologically constrained barrels with MLPs, regardless of the number of layers, leads to a substantial performance decline. This is likely due to the fact that biologically inspired recurrent spiking neural networks possess more complex temporal dynamics, enabling them to capture intricate time dependencies. And, their natural capacity to retain historical states makes them more effective for sequential tasks.
>
> Furthermore, **we evaluated the effectiveness of incorporating localist routing (With LR) vs. without localist routing (No LR) within MLP units**. The average results of three repeated experiments are shown in the table below. Overall, integrating this mechanism demonstrated better performance, particularly on the more challenging EvTouch-Containers dataset, indicating the generalizability of the localist routing mechanism.
>
> | Datasets | 2-layer MLP (No LR) | 2-layer MLP (With LR) | 3-layer MLP (No LR) | 3-layer MLP (With LR) | 4-layer MLP (No LR) | 4-layer MLP (With LR) |
> |---|:---:|:---:|:---:|:---:|:---:|:---:|
> | EvTouch-Objects | **90.74** | 88.87 | 88.193 | **88.89** | 88.43 | **88.66** |
> | EvTouch-Containers | 76.67 | **82.78** | 72.22 | **81.67** | 72.78 | **80.56** |
>
> **(c) Remove parameter sharing → independent weights Measure performance drop on EvTouch-Containers**
> Fig. 3A-B in the paper show the results of shared parameters and independent parameters on both EvTouch-Objects and EvTouch-Containers datasets. Independent parameters led to performance drops of 9.5% and 14.5%, respectively. **We will revise Fig. 3 to make the content more intuitive and clear.**
>
> ***
>
> > Q4: Does performance degrade with scaled sensor arrays? example: Simulate larger arrays (78/156 sensors) via: (a) Sensor duplication + noise (b) Zero-padding unused inputs
>
> **A4:** We agree that scaling analysis is an insightful extension with potential applications in tactile scenarios like artificial skin. While feasible in principle, this experiment has limitations: the two tasks in the paper rely on publicly available datasets (*"TactileSGNet: A Spiking Graph Neural Network for Event-Based Tactile Object Recognition," 2020*), and no real data from a scaled sensor array is available. The two alternatives you proposed, *"Sensor duplication + noise"* and *"Zero-padding unused inputs,"* can simulate signals from additional sensors, but another challenge arises—estimating the spatial coordinates of these sensors. Localist routing relies on constructing the sensor's spatial topology, which introduces additional assumptions.
>
> Returning to your original question (*"Does performance degrade with scaled sensor arrays?"*), **we believe the answer is affirmative**, even without conducting additional experiments. Adding noise without increasing useful information is expected to reduce performance. We appreciate this thoughtful suggestion and plan to explore larger-scale tactile datasets in future work.

---

> > ### Comment · Reviewer_s3by · 2025-08-03
> > **is biological complexity necessary?**
> >
> > The answers to my previous questions are fine. This study is interesting, but I noticed a potential logical inconsistency in your approach to biological constraints. You justify that the network benefit greatly from using 8 distinct excitatory neuronal subtypes and 37 synaptic pathways for "biological fidelity," yet you completely eliminate 3-4 major inhibitory subtypes—which constitute ~25% of cortical neurons—and replace them with simple negative weights.
> > This appears to be "selective biological realism" rather than genuine biological fidelity, which undermines your claims about benefits from biological constraints.
> > Two simple questions:
> > 1. What happens when you introduce at least 3 types of inhibitory interneurons (PV+, SST+, VIP+) into the network with similar detailed treatment as your 8 excitatory types?
> > 2. Have you conducted ablation studies on your 8 excitatory neuronal subtypes? What happens if you simplify to 1 or 2 subtypes instead of 8?

---

> ### Author Response · Authors · 2025-08-04
> **Logic of Using Biological Constraints**
>
> Thank you for your thoughtful comments. As you correctly noted, we did not explicitly include inhibitory interneurons in our model. The reason for this choice is that our objective was not to develop a fully detailed cortical simulation but rather to identify and integrate core principles of cortical organization that could meaningfully inspire AI design.
> Thus, the term "biological fidelity" is meant to be understood in the context of mainstream spiking neural networks or neuromorphic computing models (e.g., comparative models discussed in our paper), which commonly overlook critical biological features such as cellular diversity and microcircuit architectures.  Our selective incorporation of biological constraints, therefore, represents a deliberate balance—capturing key cortical structures while ensuring the computational efficiency and practical trainability necessary for real-world AI applications.
>
> We are aware that computational neuroscientists, such as those involved in Human Brain Project, have developed highly detailed brain models that include dendrites and various ion channels. However, such complex models are challenging to train and difficult to endow with meaningful behavioral functionalities.
>
> Next, we will address the two questions you raised point by point.
>
> ***
>
> > What happens when you introduce at least 3 types of inhibitory interneurons (PV+, SST+, VIP+) into the network with similar detailed treatment as your 8 excitatory types?
>
> This is an issue we carefully considered during the construction of our model. Early in our design process, we considered a column model that included inhibitory neurons (e.g., PV+, SST+, and VIP+). However, we ultimately opted for a more streamlined approach due to the following critical factors:
>
> 1. Excessive recurrent connections can result in vanishing gradients in deeper layers, which is an inherent limitation of gradient descent and backpropagation algorithms.
> 2. Increasing the number of neurons would require significantly more hardware resources when scaling to a multi-barrel model.
> 3. Given that inhibitory neurons mainly function to maintain network stability, we implicitly account for their inhibitory effects by allowing negative connection weights.
>
> In sum, this design reflects **a trade-off between biological fidelity and implementation efficiency**, while also underscoring the importance of developing efficient non-gradient learning methods and exploring the adaptability of biological neural networks to hardware.
>
> ***
>
> Regarding the second question:
> > Have you conducted ablation studies on your eight excitatory neuronal subtypes? What happens if you simplify to 1 or 2 subtypes instead of 8?
>
> We thank the insightful question, which is **precisely the motivation behind our experiment in Fig. 4**. By constructing a functional, trainable, and testable model, we aimed to quantify the significance of the biological constraints employed. Fig. 4B illustrates the contribution of 37 synaptic connections, with some showing a value of zero, suggesting redundancy and potential simplification. However, this does not imply that these connections are unimportant in the cortex. It is possible that our model failed to effectively differentiate their specific roles, leading to compensation by other connections with similar computational roles.
>
> Overall, this question emphasizes the core motivation behind our integration of anatomical structure into AI architectures. By constructing quantifiable and testable models, we seek to uncover which biological mechanisms are essential and use these insights to meaningfully advance bio-inspired AI design.

---

> > ### Comment · Reviewer_s3by · 2025-08-04
> >
> > Thank you for the clarification, but the response contains several logical inconsistencies that don't adequately address the core questions:
> > 1. You mention "excessive recurrent connections can result in vanishing gradients" as justification for eliminating inhibitory neurons. However, this problem applies equally to your 8 excitatory types with 37 connections. If gradient vanishing is a genuine concern, why not simplify both systems rather than selectively eliminating only inhibitory complexity?
> > 2. Regarding "increasing the number of neurons would require significantly more hardware resources." This misses the point. The question isn't about increasing neuron numbers, but about finding the optimal number of neuron types. You could maintain the same total neuron count while testing different type configurations (e.g., 2 excitatory + 2 inhibitory types vs. 8 excitatory + 0 inhibitory).
> > 3. Figure 4's finding of "redundancy" actually supports the necessity of simplification, not the current 8-type complexity. If some connections show "zero contribution," this suggests the biological complexity may be unnecessarily elaborate rather than functionally essential.
> > You haven't provided any empirical evidence comparing simplified configurations (e.g., 4 excitatory + 2 inhibitory types) against your current design. Without such ablations, the design choices appear to be arbitrary rather than principled.

---

> ### Author Response · Authors · 2025-08-05
> **Principles of Designing Models**
>
> We sincerely thank you for your valuable concerns regarding the principles behind our model design. First, we would like to emphasize that training biologically constrained models—especially those featuring extensive recurrent structures and numerous neuronal subtypes—is an exceptionally new and challenging area of research.  Successful examples of such models are currently very limited in fields such as computational neuroscience and neuromorphic computing.  As the field is in an exploratory stage, there is still no universally agreed-upon set of principles to guide such designs.  Naturally, **it would not be feasible to achieve success through arbitrary design choices**.  Indeed, throughout our research process, we have conducted numerous trials guided by two main principles: (1) established neuroscientific insights, and (2) rigorous empirical testing of functional effectiveness.
>
> We agree that our current work should be viewed as an initial exploration within this emerging research domain.  There undoubtedly remains substantial room for further refinement, optimization, and exploration. Below, we will further response to your question and elaborate on our design principles:
>
> ### 1. Prioritizing the simplification of inhibitory neurons over excitatory neurons
>
> As mentioned before, challenges such as vanishing gradients and computational cost necessitate reducing the number of recurrent connections and cells. **The choice to simplify inhibitory neurons instead of excitatory neurons is driven by their distinct functional roles in the brain.** Cortical projection neurons, primarily excitatory, form the structural and functional backbone of neural circuits and are essential for preserving the biological plausibility of network structures. In contrast, inhibitory neurons primarily regulate network stability.  We argue that their role can be effectively approximated by allowing connection weights to take negative values, making the simplification of inhibitory neurons both feasible and practical.
>
> ### 2. Progressive design strategy: starting from the comprehensive model
>
> Our model does not rely on arbitrary selection of neuron types; instead, it begins with the comprehensive configuration and systematically explores what can be simplified for practical efficiency. Early in the design process, we considered a model including eight excitatory types and eight inhibitory types, such as L2/3 PV/SST/VIP and L4 PV/SST…, rather than just focusing on three predominant inhibitory types. This represents the most comprehensive survey of the barrel cortex by neuroscientists, and an overall perspective can be obtained from "*Neuronal Circuits in Barrel Cortex for Whisker Sensory Perception, 2021*." Through iterative exploration, we simplified the inhibitory neurons to improve computational efficiency. Meanwhile, the 37 excitatory connections still represent the skeletal pathways of the barrel cortex identified by neuroscientists. Once our model demonstrated strong performance on challenging datasets, **we prioritized preserving as many biological constraints as possible to maximize fidelity**.
>
> ### 3. Balancing biological fidelity and real-world task performance
>
> Our goal is to bridge artificial neural networks and biological neural networks: artificial networks often overlook cellular structure, while biological networks lack practical functionality. Achieving convergence between the two, under current constraints, requires making trade-offs. While this necessitates sacrificing some biological precision, we ensure that our framework is quantifiable and testable, enabling us to identify the influence of biological mechanisms (as shown in Fig. 4).   Based on these results, we can iteratively refine the network architecture, as part of future work beyond the current study.
>
> We appreciate your proposal to test alternative configurations or deeper simplifications. The purpose of this study is not to propose the simplest columnar model—though we agree that it is an intriguing direction. In this paper, our focus is on the laminar structure and localist routing of the barrel cortex. We have never claimed that our current model is globally optimized，and we look forward to future research that builds upon and improves our approach.

---

> > ### Comment · Reviewer_s3by · 2025-08-05
> >
> > Thank you for the response. I acknowledge the study's novelty, but could you directly answer the remain concerns:
> > 1. Contradictory Logic: You call excitatory neurons the 'essential backbone' to justify complexity, yet Figure 4B shows several excitatory connections contribute zero performance. How are redundant connections 'essential'? You claim "excitatory neurons form the backbone while inhibitory neurons provide regulation," but fail to explain why this 'backbone' would be immune to gradient vanishing effects.
> > 2. Would a 4-excitatory + 2-inhibitory configuration perform worse than 8+0 design? Without testing simpler configurations, the design choices appear arbitrary rather than principled.

---

> > > ### Author Response · Authors · 2025-08-06
> > >
> > > Thank you for recognizing the innovation in our work. Below, we directly address the two critical questions you raised.
> > >
> > > > Q1: Contradictory Logic: You call excitatory neurons the 'essential backbone' to justify complexity, yet Figure 4B shows several excitatory connections contribute zero performance. How are redundant connections 'essential'? You claim "excitatory neurons form the backbone while inhibitory neurons provide regulation," but fail to explain why this 'backbone' would be immune to gradient vanishing effects.
> > >
> > > **A1:** We believe there is a misunderstanding regarding our use of the term "essential backbone." By this term, we refer to the complete laminar excitatory projection pathways defined in our model, including both high-impact and low-impact (potentially redundant) connections. Establishing this comprehensive structure is crucial as it provides the foundational framework needed to perform systematic analyses, such as those presented in Fig. 4. **The logic is sequential:** first defining a complete columnar structure, and then identifying essential versus non-essential components to guide future optimization and simplification.
> > >
> > > We agree in principle that **reducing either excitatory or inhibitory neuron types can decrease recurrent complexity and thus mitigate vanishing gradient issues**. However, our decision to preferentially simplify inhibitory rather than excitatory neurons was driven not purely by performance considerations but primarily by biological plausibility and interpretability. Excitatory neurons form the major functional pathways within cortical laminar circuits—pathways we consider fundamental and essential to preserve. Simplifying inhibitory neurons, which mainly serve regulatory and modulatory functions rather than primary pathway transmission, allowed us to maintain visibility into these key excitatory pathways while quickly achieving stable, trainable configurations.
> > >
> > > In summary, our selective simplification strategy was carefully chosen to balance computational feasibility, biological realism, and interpretability of cortical structures, rather than solely optimizing for maximum performance.
> > >
> > > ***
> > >
> > > > Q2: Would a 4-excitatory + 2-inhibitory configuration perform worse than 8+0 design? Without testing simpler configurations, the design choices appear arbitrary rather than principled.
> > >
> > > **A2:** During our model development, we explicitly considered alternative configurations similar to your proposed 4-excitatory + 2-inhibitory scenario. However, we quickly identified that the primary challenge involved was not solely performance but practical feasibility and biological interpretability. Given the diversity of neuronal subtypes, determining which types to retain or omit and defining their connectivity—especially the broader and denser inhibitory connections—**creates a vast combinatorial search space**. Systematically exploring and validating such extensive configuration possibilities would require significant computational resources and experimental effort, substantially exceeding the scope of this initial study. Thus, rather than arbitrary decisions, our current design represents a carefully considered initial step informed by neuroscientific principles and empirical evaluation.  We agree that extensive scope remains for systematic simplifications and optimizations in future work, which we indeed plan to pursue based on insights gained from this foundational model.

---

> > > > ### Comment · Reviewer_s3by · 2025-08-06
> > > >
> > > > Thank you for the additional responses. These questions were to assess how solid this study is, but it doesn't diminish its impressive novelty. Certainly, the work would benefit from more systematic exploration to clarify the optimal trade-off between simplicity and biological fidelity for achieving high performance.

---

### Official Review · Reviewer_qWQh · 2025-07-02

**Clarity:** 3
**Significance:** 4
**Originality:** 3
**Rating:** 5
**Confidence:** 3

**Summary:**

This paper proposed the Modular Network using Localist Topographic Expert Routing, which is inspired by Rodent Barrel Cortex . The method achieved state-of-the-art performance on challenging 3D tactile 13 object classification benchmarks, which shows the effectiveness of this method. The biologically inspired design is novel and interesting, and the experimental results have also demonstrated the effectiveness of the proposed method.

**Questions:**

Question：

Are the different modules within a functional column designed differently, or are they designed identically with each module merely corresponding to the concept of a biological structure?

**Ethical Concerns:**

["NO or VERY MINOR ethics concerns only"]

**Final Justification:**

Based on the article and the authors' responses, I consider that this work is more of a biologically inspired approach at the conceptual level. Due to the complexity of biological mechanisms and the current challenges of compatibility with existing artificial networks, I think this is a very good attempt even if it only draws on biology conceptually. Therefore, I will maintain my original rating.

**Limitations:**

Yes

**Quality:**

3

**Strengths And Weaknesses:**

Strengths:
1. The biologically inspired design is novel.

2. Routing within and between modules can effectively improve computational efficiency, and the experimental results also demonstrate advantages in performance.

3. It demonstrates how cortical principles inspire machine learning, which is interesting and illuminating.

Weakness:
1. There seems to be no further introduction on how the modules within the functional columns are designed.

2. The article mentions MoE multiple times, but there is no comparison with simple MoE. This could have better demonstrated the effectiveness of the biologically inspired routing.

---

> ### Author Rebuttal · Authors · 2025-07-29
>
> Thank you for taking the time to review our work and raising valuable concerns. We will address each point individually.
>
> > W1: There seems to be no further introduction on how the modules within the functional columns are designed.
>
> **A1:** Thank you for this valuable suggestion. The description of the internal structure of barrel columns in our paper is indeed insufficient. **In response A3 to Q1**, we have provided more detailed information on how the neural circuits are constructed, which will be added to the supplementary materials.
>
> ***
>
> > W2: The article mentions MoE multiple times, but there is no comparison with simple MoE. This could have better demonstrated the effectiveness of the biologically inspired routing.
>
> **A2:** To make the experimental results more comprehensive, **we supplemented two simple baseline MoE models for comparison:** Top-2 MoE and Dense MoE, with different numbers of experts. Each expert is a two-layer MLP with a hidden layer of 256 neurons, and a gating network allocates signals from 39 tactile sensors to the respective experts. Computations along the temporal dimension are performed independently. Each configuration was tested three times, and the best results are as follows:
>
> | Datasets |&#124;  | Top-2 MoE | |&#124; | Dense MoE | |&#124; Ours |
> |----------|:---------:|:---------:|:---------:|:---------:|:---------:|:---------:|:----:|
> |          | 16 experts | 32 experts | 48 experts | 16 experts | 32 experts | 48 experts |      |
> | EvTouch-Objects | 88.89 | 90.28 | 90.97 | 90.28 | 89.58 | 88.89 | **94.44** |
> | EvTouch-Containers | 70 | 73.33 | 73.33 | 66.67 | 75 | 70 | **86.67** |
>
> It can be observed that these two MoE baselines perform significantly worse than our multi-barrel model on both tactile datasets.  This may be due to the misalignment of the MoEs with the tactile data (our model follows a cortical one-to-one mapping) or the limited capacity of the MLP units.
>
> However, we guess that you might be more interested in how the localist routing mechanism in our model would perform when applied to mainstream large-scale MoE architectures on LLM benchmarks—a question that also greatly interests us. This poses several challenges: How should "neighboring" experts be defined? How can communication between experts be modeled? Addressing these questions may require some abstracting principles, such as "expert grouping", which would almost be another whole different work and is a promising direction for our future research.
>
> ***
>
> > Q1: Are the different modules within a functional column designed differently, or are they designed identically with each module merely corresponding to the concept of a biological structure?
>
> **A3:** Each column in our 39-barrel model shares the same structure. The column module was designed based on existing neuroscience studies and replicated 39 times to form the 39-barrel model. This replication enables parameter sharing between columns (Fig. 3). To clarify the structure of the barrel columns, the table below provides detailed information about the 8 neuron types within each barrel and **will be included in the supplementary materials**：
>
> |Abbreviation|&emsp;&emsp;&emsp;&emsp;&emsp;Scientific Name|&emsp;Type|&emsp;&ensp;Location|&emsp;References|
> |:---|:---:|:---:|:---:|:---:|
> | L2P | L2 pyramidal neurons | Excitatory | Upper Layer 2/3 | [1,2,3,4] |
> | L3P | L3 pyramidal neurons | Excitatory | Lower Layer 2/3 | [4,5,6] |
> | Pyr | Star pyramidal and pyramidal neurons | Excitatory | Layer 4 | [7,8,9,10,11] |
> | SSP | Spiny stellate pyramidal neurons | Excitatory | Layer 4 | [6,7,8,12] |
> | STP | Slender-tufted pyramidal neurons | Excitatory | Layer 5a | [13,5,14] |
> | TTP | Thick-tufted pyramidal neurons | Excitatory | Layer 5b | [15,16,17,18,19] |
> | CTP | Cortico-thalamic pyramidal neurons | Excitatory | Layer 6a | [20, 21, 22,23] |
> | CCP | Cortico-cortical pyramidal neurons | Excitatory | Layer 6a | [21, 23, 24,25] |
>
> The article "*Neuronal Circuits in Barrel Cortex for Whisker Sensory Perception, 2021*" offers an overall view on the barrel column structure. We constructed a connectivity matrix (Fig. 2A) based on the projection preferences of these neuron types reported in neuroscience studies, with connection weights automatically updated during training.
>
> ### **References:**
>
> [1] Van Brederode JF et al., Morphological and electrophysiological properties of atypically oriented layer 2 pyramidal cells of the juvenile rat neocortex, 2000.
> [2] Bureau I et al., Interdigitated paralemniscal and lemniscal path-ways in the mouse barrel cortex, 2006.
> [3] Staiger JF et al., A gradual depth-dependent change in connectivity features of supragranular pyramidal cells in rat barrel cortex, 2015.
> [4] Sermet BS et al., Pathway-, layer- and cell-type-specific thalamic input to mouse barrel cortex, 2019.
> [5] Lefort S et al., The excitatory neuronal networkof the C2 barrel column in mouse primary somatosensory cortex, 2009.
> [6] Petersen CC et al., Functionally independent columns of rat somatosensory barrel cortex revealed with voltage-sensitive dye imaging, 2001.
> [7] Egger V et al., Subcolumnar dendritic and axonal organization of spiny stellate and star pyramid neurons within a barrel in rat somatosensory cortex, 2008.
> [8] Staiger JF et al., Functional diversity of layer IV spiny neurons in rat somatosensory cortex: quantitative morphology of electrophysiologically characterized and biocytin labeled cells, 2004.
> [9] Schubert D et al., Cell type-specific circuits of cortical layer IV spiny neurons, 2003.
> [10] Brecht M et al., Dynamic representation of whisker deflection by synaptic potentials in spiny stellate and pyramidal cells in the barrels and septa of layer 4 rat somatosensory cortex, 2002.
> [11] Moore CI et al., Spatio-temporal subthreshold receptive fields in the vibrissa representation of rat primary somatosensory cortex, 1998.
> [12] Feldmeyer D et al., Reliable synaptic connections between pairs of excitatory layer 4 neurons within a single ‘barrel’ of developing rat somatosensory cortex, 1999.
> [13] Frick A et al., Monosynaptic connections between pairs of L5A pyramidal neurons in columns of juvenile rat somatosensory cortex, 2008.
> [14] Oberlaender M et al., Three-dimensional axon morphologies of individual layer 5 neurons indicate celltype-specific intracortical pathways for whisker motion and touch, 2011.
> [15] Chagnac-Amitai Y et al., Burst generating and regular spiking layer5 pyramidal neurons of rat neocortex have different morphological features, 1990.
> [16] Connors BW et al., Electrophysiological properties of neocortical neurons in vitro, 1982.
> [17] Le Be J-V et al., Morphological, electrophysiological, and synaptic properties of corticocallosal pyramidal cells in the neonatal rat neocortex, 2007.
> [18] Markram H, A network of tufted layer 5 pyramidal neurons, 1997.
> [19] Molnár Z et al., Towards the classification of subpopulations of layer V pyramidal projection neurons, 2006.
> [20] Kumar P et al., Inter- and intralaminar subcircuits of excitatory and inhibitory neurons in layer 6a of the rat barrel cortex, 2008.
> [21] Pichon F et al., Intracortical connectivity of layer VI pyramidal neurons in the somatosensory cortex of normal and barrelless mice, 2012.
> [22] Qi G et al., Dendritic target region-specific formation of synapses between excitatory layer 4 neurons and layer 6 pyramidal cells, 2016.
> [23] Zhang ZW et al., Intracortical axonal projections of lamina VI cells of the primary somatosensory cortex in the rat: a single-cell labeling study, 1997.
> [24] Narayanan RT et al., Beyond columnar organization: cell type- and target layer-specific principles of horizontal axon projection patterns in rat vibrissal cortex, 2015.
> [25] Mercer A et al., Excitatory connections made by presynaptic cortico-cortical pyramidal cells in layer 6 of the neo-cortex, 2005.

---

> > ### Comment · Reviewer_qWQh · 2025-08-05
> > **Are the different modules within a functional column designed differently？**
> >
> > Thank you for the authors' response. The authors have answered that each functional column has the same design. However, my question is whether the modules within the functional column, such as L2P, L3P, Pyr, etc., are designed individually with reference to biology or have the same design but draw on biology conceptually. Due to the complexity of biological mechanisms and the current challenges of compatibility with existing artificial networks, I think this is a very good attempt even if it only draws on biology conceptually.

---

> > > ### Author Response · Authors · 2025-08-05
> > > **Identical Spiking Neuron Model with Distinct Connectivity**
> > >
> > > Thank you for your clarification, I now get your point. **These neuron types are modeled using the same Adp LIF backbone but differ in their connectivity patterns.** Ideally, each neuron type would have distinct electrophysiological parameters; however, due to the current lack of relevant neuroscience data, they are initialized using the same spiking neuron model. But, the projection preferences of neurons are distinct, forming a distinctive topology (referred to as a "Probabilistic skeleton" in "*Probabilistic skeletons endow brain-like neural networks with innate computing capabilities, 2021*"). We mapped the connectivity matrix of these cells (Fig. 2A) based on neuroscience findings, and agree that adopting a more refined and differentiated modeling approach would be a meaningful direction for improvement.

---

### Official Review · Reviewer_FAhN · 2025-07-02

**Clarity:** 4
**Significance:** 3
**Originality:** 4
**Rating:** 5
**Confidence:** 4

**Summary:**

This paper describes a localist MoE architecture based on sensory neuroscience, specifically rodent whisker-barrel cortex. Each expert receives input from a unique sensory channel and experts communicate with their neighbors in the topology induced by the input space. Each expert is a spiking NN modeled after laminar cortex. The model shows SOTA performance on two haptic object classification tasks.

**Questions:**

I realize the ML-brain analogy is intended to go only so far, but is it plausible for cortical columns to share learnable synaptic weights? Assuming no, is there a different solution available to the brain?

For the goal of developing scalable ML systems based on this architecture, it would help to know whether the spiking dynamics and within-column recurrence are essential. Have you investigated your framework in FF analog networks?

It would also help to know the relative importance of the localist MoE architecture versus the biologically based organization within each expert (e.g., differentiated cell types and cortical layers with predefined circuitry).

My understanding of lateral connections in the brain is that they are primary inhibitory for computing local contrast. Fig 2c seems to show the opposite. What patterns do the learned inter-barrel weights show in the main experiments?

**Ethical Concerns:**

["NO or VERY MINOR ethics concerns only"]

**Final Justification:**

Novel localist MoE architecture with implications for neuroscience and SOTA performance on 3D tactile tasks. Future work is needed to scale up but the conceptual contribution is clearly above threshold.

**Limitations:**

yes

**Quality:**

3

**Strengths And Weaknesses:**

Strengths:

Novel localist MoE architecture

Impressive experimental results

Weaknesses:

It's unclear how scalable the approach is. The current implementation using spiking neurons and recurrence. The ablation study is a bit concerning in this regard as it suggests recurrence is essential to the results.

---

> ### Author Rebuttal · Authors · 2025-07-29
>
> We appreciate your insightful and valuable question. Below, we will address your concerns in detail, one by one.
>
> > Q1: I realize the ML-brain analogy is intended to go only so far, but is it plausible for cortical columns to share learnable synaptic weights? Assuming no, is there a different solution available to the brain?
>
> **A1:** This is an interesting and important question.  **We believe that the brain's parameters consist of two components:** a fixed foundational structure and fine-tuned synaptic plasticity.  The former represents genetically preserved features shaped by evolution, such as brain region organization, connectivity, and electrophysiological parameters (discussed in *"Probabilistic skeletons endow brain-like neural networks with innate computing capabilities", 2021*).  These structural features are shared among individuals, require no learning, and provide a baseline level of intelligence present at birth.  On the other hand, synaptic plasticity, such as Hebbian learning rules, enables individuals to acquire and refine their intelligence through experience, serving as fine-tuning on top of the foundational structure.
>
> In the barrel cortex, individual barrels naturally exhibit similar neural distributions and connectomes, which form the core functional structure.  While differences in whisker usage frequency may lead to variations in synaptic weights, these are supplementary adjustments rather than the primary basis for function.  Therefore, we believe the structure of barrel columns largely adheres to the principle of modular reuse.
>
> What makes this particularly interesting is its alignment with the current development of large models in artificial intelligence.  Researchers often use a limited set of pre-trained model parameters as a foundation and then fine-tune them to develop expert models in different domains.  Pre-trained parameters resemble genetic information, while post-training processes can even reveal a "lineage" of large models based on the statistics of shared parameters.  From this perspective, studying the convergent evolution of natural and artificial intelligence represents an exciting future direction.
>
> ***
>
> > Q2: For the goal of developing scalable ML systems based on this architecture, it would help to know whether the spiking dynamics and within-column recurrence are essential. Have you investigated your framework in FF analog networks?
>
> **A2:** We agree that ablation experiments for biological constraints are important, and we will provide additional experimental results in detail in our response A3 for the next question. Here, we would like to focus specifically on the Forward-Forward network. As you pointed out, there is currently no definitive evidence for end-to-end backpropagation occurring in the brain. However, it remains a practical and stable learning algorithm, which is why it is used in our model. In fact, one of the main reasons we avoid explicitly defining inhibitory neurons is that excessive recurrent connections could result in vanishing gradients in distant layers.
>
> Brain-inspired local learning rules, such as Hebbian learning and Spike-Timing-Dependent Plasticity (STDP), offer significant advantages in terms of efficiency and cost. However, in their current form, they are less practical than backpropagation for achieving comparable performance. Therefore, we continue to use gradient descent and backpropagation to balance the biological plausibility of our model with its behavioral functionality. We believe integrating brain-inspired learning rules is an important research direction, and it relies on the development of more effective local learning algorithms in the future.
>
> ***
>
> > Q3: It would also help to know the relative importance of the localist MoE architecture versus the biologically based organization within each expert (e.g., differentiated cell types and cortical layers with predefined circuitry).
>
> **A3:** We conducted three supplementary ablation experiments to evaluate the relative importance of biological constraints. First, **we replaced the horizontal localist routing in the model with random connections** while keeping the number of neighbors fixed at 4. Each configuration was trained three times, and the best results are shown in the table below:
>
> | Datasets | &#124; Random connection &#124; | Localist routing |
> |:---|:---:|:---:|
> |EvTouch-Objects|90.28|**94.44**|
> |EvTouch-Containers|83.33|**86.67**|
>
> As shown, replacing localist routing with random connections led to performance drops of approximately 4.2% and 3.3%, respectively, highlighting the advantages of localist routing for these tactile perception tasks.  This is likely because object textures often exhibit spatially structured patterns, and localist routing preserves this neighborhood information, facilitating the integration of spatially correlated signals for better classification.  In contrast, random connections disrupt this spatial organization, introducing noise and reducing the model’s ability to capture meaningful relationships between neighboring signals.  By leveraging the inherent spatial coherence of tactile data, localist routing aligns more closely with the structured nature of sensory inputs, making it more effective for these tasks.
>
> Next,  **we replaced each barrel in the multi-barrel model with an MLP** of varying depths while keeping other configurations unchanged.  The MLPs shared parameters, with each hidden layer containing 256 neurons.  Each configuration was trained three times, and the best results are shown below:
>
> | Datasets | &#124; 2-layer MLP &#124; | 3-layer MLP &#124; | 4-layer MLP &#124; | Biological barrel |
> |:---|:---:|:---:|:---:|:---:|
> |EvTouch-Objects|89.58|88.89|88.89|**94.44**|
> |EvTouch-Containers|83.33|83.33|81.67|**86.67**|
>
> It is evident that replacing biologically constrained barrels with MLPs, regardless of the number of layers, results in a significant performance drop.  This is likely because biologically inspired recurrent spiking neural networks exhibit richer temporal dynamics, allowing them to capture complex time dependencies.  Additionally, their inherent ability to store historical states makes them better suited for sequential tasks.  However, understanding how the biological structure within a barrel (e.g., laminar structure and neuronal distributions) aligns with tactile tasks requires deeper insights from neuroscience.
>
> Furthermore, **we tested whether localist routing remains effective within the MLP units** (With Localist Routing vs. No Localist Routing). The average results of three repeated experiments are presented in the table below. Overall, incorporating localist routing achieved better performance, particularly on the more challenging EvTouch-Containers dataset, indicating the generalizability of our model.
>
> | Datasets | 2-layer MLP (No LR) | 2-layer MLP (With LR) | 3-layer MLP (No LR) | 3-layer MLP (With LR) | 4-layer MLP (No LR) | 4-layer MLP (With LR) |
> |-----------------|:-----------:|:-----------:|:-----------:|:-----------:|:-----------:|:-----------:|
> | EvTouch-Objects | **90.74** | 88.87 | 88.193 | **88.89** | 88.43 | **88.66** |
> | EvTouch-Containers | 76.67 | **82.78** | 72.22 | **81.67** | 72.78 | **80.56** |
>
> ***
>
> > Q4: My understanding of lateral connections in the brain is that they are primary inhibitory for computing local contrast. Fig 2c seems to show the opposite. What patterns do the learned inter-barrel weights show in the main experiments?
>
> **A4:** Thank you for pointing out this potentially confusing aspect. While lateral inhibition is widely recognized, it is important to note that the **horizontal connections in the barrel cortex include both inhibitory and excitatory types** (clearly illustrated in *Neuronal Circuits in Barrel Cortex for Whisker Sensory Perception, 2021*).
>
> In reality, rodent whiskers facilitate two forms of tactile perception: active (referred to as "whisking") and passive (referred to as "touch") [1, 2, 3]. During whisking, muscles actively move the whiskers to sweep over surrounding objects, triggering brief lateral inhibition to enhance precise whisker localization. Neural responses in this case are smaller and more localized. In contrast, during passive touch, external stimuli to a single barrel elicit larger and more widespread neural responses, which is illustrated in Fig. 2C-D in our paper.
>
> For machine learning classifiers, **excitatory inter-barrel currents can similarly enhance neural activity differences** (Fig. 5E). By selectively modulating inter-barrel currents, they amplify relative activity contrasts between barrels, thereby encouraging competition. When neural activity across barrels is highly uniform, it suggests the encoding of redundant information. Thus, even without explicitly defining lateral inhibition, the inter-barrel currents in our model, combined with machine learning techniques, can selectively amplify perceptual differences to improve representation diversity.
>
> ### **References:**
>
> [1] Carl C.H. Petersen and Sylvain Crochet, Synaptic Computation and Sensory Processing in Neocortical Layer 2/3, 2013.
> [2] Carl C. H. Petersen, Sensorimotor processing in the rodent barrel cortex, 2019.
> [3] Anton Sumser et al., Active and passive touch are differentially represented in the mouse somatosensory thalamus, 2025.

---

> > ### Comment · Reviewer_FAhN · 2025-08-05
> >
> > Thanks for the thoughtful replies. I think you've done a thorough job replying to all the reviewers' comments.
> >
> > Reviewer NTKp makes a good case about the contributions. The paper is largely exploratory but I think that’s okay given its novelty.
> >
> > The comparisons to MoE may be exaggerated (i.e., the present approach is different) because MoE assigns different input examples to different experts, whereas the present model assigns different parts of each input. In fact it’s more like a CNN because each column is a copy of the same kernel acting on a different region of the input.
> >
> > I agree with Reviewer s3by’s main point about identifying which biological constraints are important to the results. Along the same lines I should clarify that in my Q2 I intended FF to mean feed-forward (vs recurrent, not a reference to learning algorithms). I’m mainly interested in whether the recurrence and more importantly the spiking neurons (vs analog) are needed for your results.

---

> > > ### Author Response · Authors · 2025-08-05
> > >
> > > We sincerely appreciate your recognition of the novelty of our work, and all the reviewers' comments have further improved its quality. Below, we would like to respond to the new issues you have raised point by point.
> > >
> > > > Q5: The comparisons to MoE may be exaggerated (i.e., the present approach is different) because MoE assigns different input examples to different experts, whereas the present model assigns different parts of each input.
> > >
> > > **A5:** We agree that our model assigns different parts of the input to distinct modules;  however, this approach is consistent with the principles of mainstream MoEs.  In MoE, different tokens are assigned to specific experts, where tokens are themselves components of a sentence, and the sentence as a whole represents the input.  Similarly, our model decomposes the input and distributes its components among modules, aligning with this concept.
> > >
> > > ***
> > >
> > > > Q6: In fact it’s more like a CNN because each column is a copy of the same kernel acting on a different region of the input.
> > >
> > > **A6**: This is an insightful observation. We agree that our multi-barrel model with shared parameters bears certain similarities to CNNs. However, there are several fundamental differences:
> > >
> > > 1. CNN kernels have a fixed, regular size, whereas our localist routing is dynamically adjusted based on a gating network.
> > > 2. CNNs progressively aggregate features across layers to enhance global dependencies, whereas in our column module, all layers (L2–L6) consistently adhere to the constraint of localist perception.
> > > 3. From a feature extraction perspective, our model is more inclined to preserve local differences, making it well-suited for modular tasks such as tactile processing.
> > > 4. Vertical and horizontal currents is coupled with the nonlinear dynamics of biological neurons, resulting in temporal dependencies and richer behaviors such as local collaboration and competition.
> > >
> > > Moreover, your question has inspired us to **explore a brain-like semi-parameter sharing mechanism**. On one hand, it could maintain a shared foundational structure to reduce parameters, and on the other hand, allow individual modules to compute gradients based on local inputs. This could be an intriguing direction for building distributed expert systems.
> > >
> > > ***
> > >
> > > > Q7: I agree with Reviewer s3by’s main point about identifying which biological constraints are important to the results. Along the same lines I should clarify that in my Q2 I intended FF to mean feed-forward (vs recurrent, not a reference to learning algorithms). I’m mainly interested in whether the recurrence and more importantly the spiking neurons (vs analog) are needed for your results.
> > >
> > > **A7:** We regret the misunderstanding of your question. It seems, however, that relevant experimental results were provided **in the response to Q3 (A3)**. In that response, we tested replacing the biological barrels (recurrent network with spiking neurons) with MLPs (feedforward network with analog neurons) of varying depths. The results indicated a notable decline in performance. For clarity, we report the relevant results again below:
> > >
> > > |Datasets	| &#124; 2-layer MLP |&#124;	3-layer MLP |&#124; 4-layer MLP |&#124; Biological barrel |
> > > |:---|:---:|:---:|:---:|:---:|
> > > |EvTouch-Objects|	89.58|	88.89|	88.89|	**94.44**|
> > > |EvTouch-Containers|	83.33|	83.33|	81.67|	**86.67**|
> > >
> > > Additionally, we would like to highlight that the use of spiking neurons serves as an interface between artificial models and biological neural networks, thereby bridging neuroscience research on neural activity, such as analyzing neural dynamics and other metrics. This enables the model not only as a tool for evaluating task performance but also as a platform for testing and exploring neuroscientific insights.

---

### Official Review · Reviewer_NTKp · 2025-07-03

**Clarity:** 3
**Significance:** 1
**Originality:** 2
**Rating:** 2
**Confidence:** 3

**Summary:**

- This paper proposes a barrel cortex-inspired modular neural network architecture for sensorimotor processing
- Each module corresponds to a cortical column and input sensory signals are routed exclusively to their specific columns/modules.
- The authors achieve state-of-the-art performance on tactile object classification while reducing parameters by 97% through columnar weight sharing

**Questions:**

- What audience do the authors think will resonate with this work and how can it be built upon?

**Ethical Concerns:**

["NO or VERY MINOR ethics concerns only"]

**Final Justification:**

Overall, I would still like the authors to down-tone their claims regarding machine learning advances made by their work. I will maintain my score.

**Limitations:**

Yes

**Quality:**

3

**Strengths And Weaknesses:**

strengths:
- Well explained biologically-inspired architecture that faithfully replicates established neuroscientific principles
- Achieves high parameter efficiency while maintaining competitive performance

weaknesses:
- The authors claim that this work “provides a step toward next-generation expert systems that bridge neuroscience and artificial intelligence”, but it is not clear to me how this work really pushes forward either community. More performant and lower parameter count models could be built if there were no biological constraints as there are numerous non-biological ways of reducing model parameter counts. From the neuroscience perspective mouse barrel cortex organization is already well understood and the authors don’t offer any suggestion on how this new model could be used to explore testable hypotheses of neuroscientific interest.

---

> ### Author Rebuttal · Authors · 2025-07-29
>
> We sincerely thank you for your feedback on the contributions of our work. Below, we would like to respond to each of your concerns point by point.
>
> > Weaknesses: The authors claim that this work “provides a step toward next-generation expert systems that bridge neuroscience and artificial intelligence”, but it is not clear to me how this work really pushes forward either community. More performant and lower parameter count models could be built if there were no biological constraints as there are numerous non-biological ways of reducing model parameter counts. From the neuroscience perspective mouse barrel cortex organization is already well understood and the authors don’t offer any suggestion on how this new model could be used to explore testable hypotheses of neuroscientific interest.
>
> **A1:** We appreciate your concern about how our work contributes to both fields, and we provide the following points of clarification.
>
> **From an AI perspective,** our work offers more than a performance improvement or parameter reduction—it fundamentally rethinks model design by drawing inspiration from brain architecture. As one of the other reviewers insightfully noted, “*It demonstrates how cortical principles inspire machine learning, which is interesting and illuminating.*” We agree that non-biological approaches can reduce parameters, but our aim is to leverage principles from the most efficient intelligent system we know—the brain—to guide AI design.
>
> Specifically, our study highlights the differences between AI expert systems, such as the MoE, and cortical expert systems like the barrel cortex. Unlike MoE, which emphasizes task allocation by routing tokens to the appropriate expert, the barrel cortex features innate one-to-one mappings and emphasizes collaboration and competition among modules. Our work explores these collaborative processes, applying them to real-world tactile tasks.
>
> While achieving SOTA performance on two challenging tactile datasets, our model’s primary goal is not to compete with unconstrained AI models purely on performance but to introduce a paradigm in designing AI systems inspired by biologically evolved structures.  Examples from bio-inspired computing already underscore the value of such an approach, including convolutional neural networks [1, 2], which were inspired by the cat’s visual cortex, and spiking neural networks [3, 4], which emulate the spiking behavior of biological neurons.  This is not the only direction for better AI design, but it is certainly a rational and promising one.
>
> **From a neuroscience perspective,** we agree that the anatomical organization of the mouse barrel cortex is well studied. However, there is a gap in translating these anatomical insights into a **functional, trainable computational model**. To our knowledge, among the six existing barrel cortex models listed in ModelDB [5, 6, 7, 8, 9, 10], none incorporate a modular multi-barrel columnar structure or support end-to-end training on behavioral tasks. Our work is the first to integrate the barrel cortex’s columnar organization and dynamics into a trainable model that balances biological plausibility with practical functionality in tactile perception tasks.
>
> This integration allows us to propose testable computational mechanisms inspired by cortical structure and dynamics. For instance, our model enables us to: (1) Examine how reusing the same module affects learning efficiency (Fig. 3 in our paper); (2) Quantify the contribution of individual neural microcircuits to overall tactile perception (Fig. 4). (3) Investigate the effect of lateral currents on network dynamics (Fig. 5). Each of these findings can be viewed as a testable prediction or hypothesis about how the real barrel cortex processes tactile information.
>
> In summary, our aim is not only to increase the practical value of biologically-inspired models, but also to leverage modern AI techniques to turn them into quantifiable frameworks that can bridge to neuroscience.  By doing so, we take a step toward next-generation expert systems that truly bridge neuroscience and AI: benefiting AI through new design principles and benefiting neuroscience through models that generate experimentally testable insights.
>
> ***
>
> > Questions: What audience do the authors think will resonate with this work and how can it be built upon?
>
> **A2:** In terms of the audience who may resonate with this work, we believe there are several key groups.
>
> 1. **Computational Neuroscientists:** Our work offers an approach of how biological realism can be combined with functional performance. Rather than focusing on anatomy alone, we demonstrate a way to incorporate cortical structures into a working model that performs real tasks.  This balance of biological plausibility and functional utility could inspire similar integrative modeling efforts for other brain regions.  As neuroscience data and knowledge continue to grow, there is a need for unified models that can both explain and predict brain function.  We hope our approach encourages the community to develop more such models that are biologically grounded yet behaviorally relevant.
>
> 2. **Neuromorphic Computing Researchers:** Built on a spiking neural network backbone, our model is well-suited for deployment on neuromorphic hardware [11], which could drastically improve energy efficiency for tactile processing. Moreover, by directly instantiating principles from sensory cortex, our approach offers a ready-made architecture for advanced tactile perception. This is especially timely given the rising interest in tactile intelligent systems and robotics [12, 13]. Our model could serve as a starting point for implementing brain-like tactile processing on neuromorphic chips, and further research can explore optimizations (e.g., sparse computing, on-chip learning) that capitalize on its brain-inspired structure.
>
> 3. **MoE Researchers:** As mentioned in the paper and response A1, current MoE research primarily focuses on task allocation—how to appropriately divide tokens and assign them to corresponding experts at the right level.  Our work highlights that the brain's natural expert systems (barrel cortex) exhibit distinct mechanisms of collaboration and competition.  These differences could inspire novel MoE architectures, such as exploring inter-expert collaboration to reduce information redundancy or leveraging inter-regional communication in the brain to handle multimodal tasks.  Some recent MoE studies [14] have started focusing on expert collaboration, and understanding how the brain divides and coordinates tasks could provide valuable insights for these efforts.
>
> We envision that researchers in these areas can build upon our model in various ways: by extending it to other sensory modalities, by increasing its biological detail (if their goal is a more accurate brain simulation), or by abstracting it further for general AI tasks.  We are excited to see how the community will take these ideas forward and we are happy to engage in a dialogue to support such efforts.
>
> ### **References**
>
> [1] Yann Lecun et al., “Gradient-Based Learning Applied to Document Recognition”, 1998.
> [2] D. H. Hubel and T. N. Wiesel, “Receptive fields, binocular interaction, and functional architecture in the cat’s visual cortex”, 1962.
> [3] Wolfgang Maass, “Networks of spiking neurons: The third generation of neural network models”, 1997.
> [4] Kaushik Roy et al., “Towards spike-based machine intelligence with neuromorphic computing”, 2019.
> [5] Lavzin et al., “Nonlinear dendritic processing determines angular tuning of barrel cortex neurons in vivo, 2012.
> [6] Kremer et al., “Late Emergence of the Vibrissa Direction Selectivity Map in the Rat Barrel Cortex”, 2011.
> [7] Diego et al.,” Mechanisms underlying a thalamocortical transformation during active tactile sensation”, 2017.
> [8] Argaman et al.,” Does Layer 4 in the Barrel Cortex Function as a Balanced Circuit when Responding to Whisker Movements? ”, 2017.
> [9] Domanski et al., “Cellular and synaptic phenotypes lead to disrupted information processing in Fmr1-KO mouse layer 4 barrel cortex ”, 2019.
> [10] Huang et al., “Cortical Representation of Touch in Silico”, 2022.
> [11] Eunhye Baek et al., “Neuromorphic dendritic network computation with silent synapses for visual motion perception”, 2024.
> [12] Meng Qi et al., “Self-powered artificial vibrissal system with anemotaxis behavior”, 2025.
> [13] Zihang Zhao et al., “Embedding high-resolution touch across robotic hands enables adaptive human-like grasping”, 2025.
> [14] Zhang et al.,  "Advancing MoE Efficiency: A Collaboration-Constrained Routing (C2R) Strategy for Better Expert Parallelism Design", 2025.

---

> > ### Comment · Reviewer_NTKp · 2025-08-09
> > **rebuttal**
> >
> > Thanks for the thoughtful rebuttal. I remain unconvinced of the significance of this works contribution to a machine learning audience, although I see the the value for neuroscience. In many ways our current deep learning models have far surpassed the abilities of the mouse brain. The argument that an architecture design is promising and that ML practitioners will want to build on it just because it is brain inspired, despite the fact that it is designed "not to compete with unconstrained AI models purely on performance", is not convincing to me.
> >
> > Overall I believe this paper contributes to building a better brain model, but I will maintain my score because I disagree with the overall framing of the work as "increasing the practical value of biologically-inspired models" and over claiming of the works contribution to AI.

---

> > > ### Author Response · Authors · 2025-08-09
> > >
> > > We appreciate your reply. We had hoped for a more engaged dialogue, but unfortunately, your response arrived **just an hour before the deadline**.
> > >
> > > We respectfully but firmly disagree with your framing of our work and the narrow view it reflects. By dismissing architectures that do not explicitly aim to outperform current deep learning models, you imply that any biologically grounded design lacking competitive benchmark metrics is irrelevant to the machine learning community. This line of thinking risks repeating the exact historical misjudgment that marginalized neural networks in the 1990s—only to see them later become the foundation of today’s AI.
> > >
> > > We agree that deep learning has made extraordinary advances. However, even today’s most advanced systems (e.g., ChatGPT-5) remain far from matching basic biological organisms in key capabilities: **generalization, sample efficiency, robustness to perturbation, and real-world sensory-motor integration**. Consider, for example, the zebrafish: a tiny organism with only ~100,000 neurons, yet capable of robustly navigating and interacting with complex underwater environments. To date, no deep learning model—not even large language models—can replicate this level of embodied general intelligence. This growing awareness has fueled renewed interest in neuromorphic and brain-inspired computation [1, 2, 3]. Dismissing such directions simply because they do not optimize for leaderboard benchmarks overlooks the increasingly interdisciplinary trajectory of our field.
> > >
> > > Our model—while grounded in cortical anatomy—achieves state-of-the-art results on two challenging public tactile benchmarks. This is not a simulation; it is a testable, trainable, scalable architecture that introduces functional insights relevant to ML. **You did not point to a single technical flaw, nor did you offer constructive engagement during the rebuttal process. Rejecting a paper on the sole basis of philosophical disagreement while ignoring demonstrated empirical performance does not serve the scientific process well.**
> > >
> > > We stand by our framing: this work increases the practical value of biologically inspired models and offers actionable insights for the ML community. We regret that the review process allowed limited room for substantive exchange.
> > >
> > > ### **References**
> > >
> > > [1] Roy et al., Towards spike-based machine intelligence with neuromorphic computing., Nature 2019.
> > > [2] Zhang et al., The development of general-purpose brain-inspired computing., Nature Electronics 2024.
> > > [3] Kudithipudi et al., Neuromorphic computing at scale., Nature 2025.

---

### Note · Authors · 2025-08-12

We thank the reviewers for your time and thoughtful feedback. This paper engages a timely and important direction in AI research. With the release of GPT-5, a growing body of scholarship has noted diminishing returns from scaling alone. Although contemporary LLMs achieve impressive results on benchmark datasets, they still fall short of the adaptive general intelligence exhibited even by simple biological organisms—particularly in robust generalization, sample efficiency, and real-world sensorimotor competence. **This observation raises a foundational question:** should AGI be defined primarily by superior performance on ever-larger curated datasets, or by the capacity to learn and adapt in open-ended physical environments? Several leading researchers argue that, despite substantial investment, scaling by itself has not produced AGI and that alternative research trajectories are needed. **We share this view:** biologically grounded models may better address the demands of embodied general intelligence. Our work contributes a trainable, testable framework that systematically integrates such principles into modern ML.

### **Responses to key questions:**
**1. Comparison to standard MoEs.** As suggested, we benchmarked against Top-2 MoE and dense MoE with matched capacity; our model outperforms both.
**2. Ablations.** Replacing localist routing with random connection, or substituting biological columns with feed-forward MLP modules, degrades performance; notably, localist routing also helps the MLP variant.
**3. Column design clarity.** We will expand neural subtype details. All subtypes share Adp LIF backbone but differ in connectivity; we will add rationale for selecting these subtypes.
**4. ML contribution.** We clarify positioning within neuromorphic computing and brain-inspired ML, achieving SOTA results on two real-world tactile datasets.

We agree that further systematic exploration is warranted (e.g., trade-offs between simplicity and fidelity, larger sensor arrays). We appreciate the reviewers’ guidance and believe the revisions will address the major concerns.

---

### Decision · Program_Chairs · 2025-09-17

**Decision:**

Accept (poster)

**Comment:**

Inspired by the rodent barrel cortex, this work introduces a modular neural network where sensory inputs are routed to specialized, locally-connected expert modules that mimic cortical columns.  This model achieves strong performance on tactile classification tasks with few parameters, and improved training stability.

The reviewers agree that this is interesting work at the intersection of neuroscience and machine learning. Reviewer NTKp raised some concerns regarding overclaiming the importance of the current findings with respect to state-of-the-art artificial neural networks. Given the current limited evaluation of the model (which is appropriate for such a first paper), I would ask the authors to manage this line carefully when writing the final version of their paper.